# The biological role of local and global fMRI BOLD signal variability in multiscale human brain organization

Giulia Baracchini [1] ✉, Yigu Zhou[1], Jason da Silva Castanheira [1], Justine Y. Hansen [1], Can Fenerci[2], Roni Setton [3], Jenny R. Rieck [4], Gary R. Turner[5], Cheryl L. Grady[6,7], Bratislav Misic [1], Jason S. Nomi[8], Lucina Q. Uddin[8] & R. Nathan Spreng [1] ✉

Variability drives the organization and behavior of complex systems, including the human brain. Understanding the variability of brain signals is thus necessary to broaden our window into brain function and behavior. Few empirical investigations of macroscale brain signal variability have been undertaken, given the difficulty in separating biological sources of variance from artefactual noise. Here, we characterize the temporal variability of the most predominant macroscale brain signal, the fMRI BOLD signal, and systematically investigate its statistical, topographical, and neurobiological properties. We contrast fMRI acquisition protocols, and integrate across histology, microstructure, transcriptomics, neurotransmitter receptor and metabolic data, fMRI static connectivity, and empirical and simulated magnetoencephalography data. We show that BOLD signal variability represents a spatially heterogeneous, central property of multi-scale multi-modal brain organization, distinct from noise. Our work establishes the biological relevance of BOLD signal variability and provides a lens on brain stochasticity across spatial and temporal scales.

Variability is ubiquitous in our environment. It is a crucial feature of complex ecological and biological systems[1]: from day-to-day climate variability driving climate change and climate mitigation efforts[2,3], to heart rate variability serving as a clinical tool to predict overall cardiovascular health and mortality[4]. Acting as a catalyst for system adaptability, variability determines the organization of complex systems, defines their spatiotemporal properties and guides their behavior[5]. Gaining insights into the variability of a system may thus unlock a deeper understanding of the system from which variability emerges.

The human brain is a complex system. From the processing of incoming information to the generation of motor outputs, variability is present at all levels of the central nervous system[6,7]. Brain activity varies across multiple timescales, from milliseconds to years, and these temporal changes can be observed across multiple spatial scales, from regions to networks[8,9]. Despite its pervasiveness, there continues to be resistance to exploring variability in human cognitive neuroscience. Human behavior has been shown to be stochastic[7,10,11], and computational models have operationalized the brain as a complex dynamical system[12–16], yet corresponding empirical neuroimaging research still lags behind. Most functional MRI (fMRI) investigations are centered on static methodological approaches to brain function. Given the richness of information present in the fMRI blood oxygen level dependent

[1]Montreal Neurological Institute, Department of Neurology and Neurosurgery, McGill University, Montreal, QC, Canada. [2]Department of Psychology, McGill University, Montreal, QC, Canada. [3]Department of Psychology, Harvard University, Cambridge, MA, USA. [4]Health Canada, Ottawa, ON, Canada. [5]Department of Psychology, York University, Toronto, ON, Canada. [6]Rotman Research Institute at Baycrest, Toronto, ON, Canada. [7]Department of Psychiatry and Psychology, University of Toronto, Toronto, ON, Canada. [8]Department of Psychiatry and Biobehavioral Sciences, University of California, Los Angeles, Los Angeles, CA, USA. ✉e-mail: giulia.baracchini@mail.mcgill.ca; nathan.spreng@mcgill.ca

(BOLD) signal, disentangling intrinsic biological sources of signal variance from extrinsic artefactual sources arising from the scanner has historically been a challenge[17]. fMRI is one of the most widely used and clinically tractable tools for the exploration of macroscale brain function. These realities position a systematic evaluation of BOLD signal variability as a research imperative, necessary to broaden our window into human brain function.

Thus far, BOLD signal variability has been primarily studied in relation to behavior, cognition, development, and clinical status[18-24]. Demonstrating such associations does not, however, directly establish BOLD signal variability's biological role in whole-brain organization. As such, it remains unclear whether and to what extent BOLD signal variability investigations are capturing meaningful neural signatures or system noise. To this end, a few recent studies have begun to characterize BOLD variability's neurobiological underpinnings, yet these investigations are typically restricted to comparisons across age and focus on single, mostly macroscale, neurobiological features[25-29]. A broader and more integrated assessment of BOLD signal variability in younger samples is therefore needed, one that examines its statistical, topographical and neuronal properties and systematically relates BOLD signal variability to a wider array of neurobiological features. Evidence in favor of taking such an integrative multi-scale approach to BOLD signal variability comes from multiple lines of empirical and computational work collectively demonstrating the role of cellular, molecular, genetic, and metabolic factors in shaping local and global hemodynamic signals[30-34].

First, BOLD signal variability is a statistical approximation of macroscale brain signal dynamics thus, both the estimation method and the properties of the fMRI data from which it is estimated will impact outcomes. Second, human brain function is organized along hierarchical modules, ranging from local functional units to global functional networks[35,36], with heterogeneous topographies. Such organization enables information to be both segregated (local) and integrated (global) across units, subserving different neural and cognitive functions—as observed in various measures, such as mean functional activation, functional connectivity, computational, and BOLD signal complexity studies[37-42]. Along these lines, BOLD signal variability must therefore also be understood within a local-global framework[43,44] that considers both the methodology and analysis level of interest. While previous fMRI reports have investigated the local-global nature of BOLD signal variability, they have generally focused on static global measures of brain organization (i.e., the correlation between local mean BOLD signals over time, functional connectivity[45]) and more local characterizations of variability on a single scale (e.g., voxel or region). The temporal variability of individual regions (local BOLD signal variability) must thus be evaluated alongside the dynamic coordination of functional neural units (dynamic functional connectivity, here global BOLD signal variability). If biologically relevant, local and global BOLD signal variability should present spatially heterogeneous topographies that recapitulate known neurobiological processes that unfold across spatial scales. Third, BOLD signal variability should reflect aspects of macroscale neuronal signals that occur at finer temporal scales and are captured by neuroscientific modalities with greater temporal resolution than fMRI, such as electrophysiology.

In this study, we defined measures of local and global BOLD signal variability as the regional moment-to-moment change in BOLD signal intensity between successive timepoints and as the similarity in interregional functional connectivity over time. To determine robust results dissociable from noise, we leveraged multiple state-of-the-art openly available fMRI datasets and assessed the validity and reliability of local and global BOLD variability across samples. To understand the spatial organization and biological properties of local and global BOLD variability, inspired by recent multiscale fMRI investigations[32,46-49], we examined their topography within each fMRI dataset and interrogated associations with open-source data, including ex_vivo histology[50] and

in_vivo microstructure[51], transcriptomics[52,53], PET-derived neurotransmitter receptor and metabolic information[47], and fMRI static connectivity data[54,55]. Finally, we leveraged magnetoencephalography (MEG) data and naturalistic electrophysiological simulations to mechanistically understand the temporal and neuronal properties of local multimodal signal variability. We found that measures of BOLD signal variability exhibited spatially heterogeneous topographies, were embedded within multi-scale brain organization, and were rooted in electrophysiological processes. Together, our work establishes local and global BOLD signal variability as biologically relevant, central features of multi-scale, multi-modal brain organization, distinct from noise. Our study represents a step forward towards understanding brain stochasticity across spatial and temporal scales.

## Results
### Quantification of local and global BOLD signal variability

We first sought to identify robust regional metrics of local and global BOLD signal variability. By local and global BOLD signal variability, we refer to changes in the dynamic properties of single timeseries (local) and of pairs of timeseries (global). The former case is what is traditionally referred to as "BOLD signal variability" in the literature, and the latter as "dynamic functional connectivity". In using the general term "variability" for both measures, we aimed at conceptually juxtaposing two lines of research that have traditionally been treated independently but recently shown to be inherently intertwined[18,44]. In doing so, our work established the foundation for future studies to quantitatively compare them. For all analyses, brain regions were defined using the Schaefer 200 regions-17 networks parcellation solution[56].

Local BOLD signal variability was calculated by taking the root mean squared successive difference (rMSSD) of each normalized regional timeseries. rMSSD quantifies moment-to-moment changes in the BOLD signal by measuring the mean of the squared differences in signal intensity between successive timepoints[57]. Greater local BOLD variability, therefore, is present in regions with greater rMSSD (Fig. 1A).

Global BOLD signal variability was estimated as dynamic functional connectivity using covSTATIS. As a three-way extension of multidimensional scaling and principal component analysis, covSTATIS is a multi-table method that linearly combines multiple similarity matrices (i.e., correlation/covariance tables) and uses eigenvalue decomposition to identify predominant structured patterns both at the group and individual level[58-60]. In our case, we used a combination of sliding window approaches and covSTATIS to examine, for each individual, the similarity of their windowed functional connectivity matrices over time and derive individual-level estimates of regional functional connectivity dynamics. Unlike most conventional fMRI dynamic connectivity methods[61] that operate on coarser brain dynamics estimates, covSTATIS defines finer regional measures of global dynamics and does so with minimal user-dependent input, as it side-steps commonly adopted clustering approaches that have traditionally led to fractionated, study-specific definitions of dynamic functional connectivity[62]. For more information about covSTATIS and its implementation, we refer the reader to our dedicated recent publication[60].

After partitioning each regional timeseries into equally sized windows via a sliding window approach[63,64] (see "Methods" for details), we derived, for each window, functional connectivity measures for each region pair, as their product-to-moment correlation across timepoints (Fig. 1B *step* 1). This procedure resulted in $N$x$N$x$T$ functional connectivity data tables for each individual, where $N$ is the number of regions and $T$ the number of windows. We next assessed, via the $R_v$ similarity coefficient[65], the similarity across all data tables across all individuals (Fig. 1B *step* 2). We then calculated their weighted average (a $N$x$N$ data table), to obtain a group compromise space, where regional connections more similar across time and individuals were given a higher weight, since they were most represented in the sample (Fig. 1B *step* 3). We submitted the group compromise space to

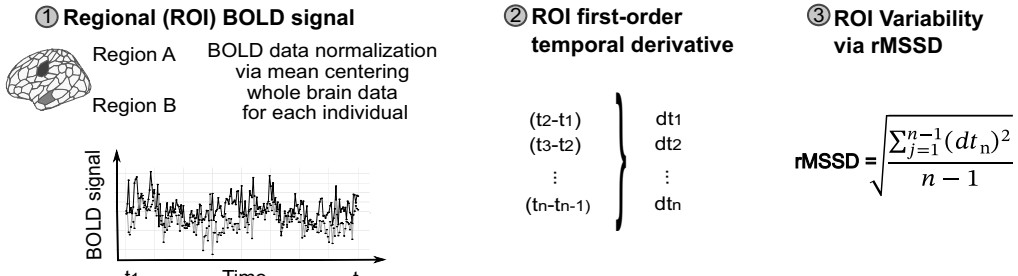

**A** **Local fMRI BOLD variability: regional BOLD signal variability via rMSSD**

① **Regional (ROI) BOLD signal**

② **ROI first-order temporal derivative**

③ **ROI Variability via rMSSD**

$$rMSSD = \sqrt{\frac{\sum_{j=1}^{n-1}(dt_n)^2}{n-1}}$$

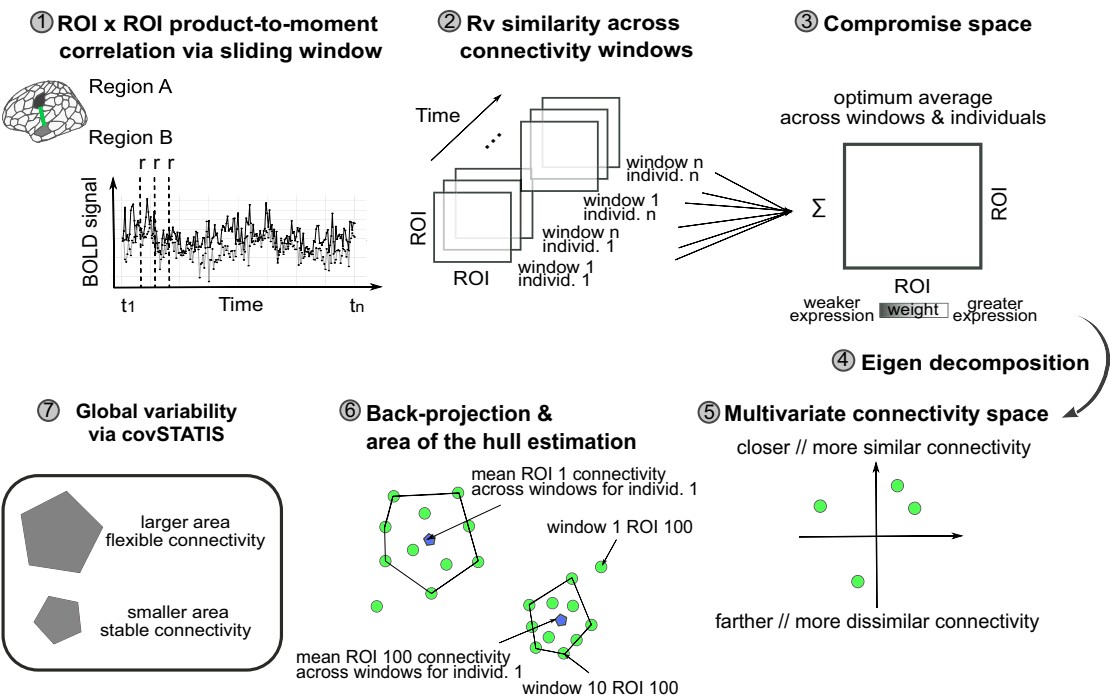

**B** **Global fMRI BOLD variability: regional dynamic functional connectivity via covSTATIS**

① **ROI x ROI product-to-moment correlation via sliding window**

② **Rv similarity across connectivity windows**

③ **Compromise space**

④ **Eigen decomposition**

⑤ **Multivariate connectivity space**

⑥ **Back-projection & area of the hull estimation**

⑦ **Global variability via covSTATIS**

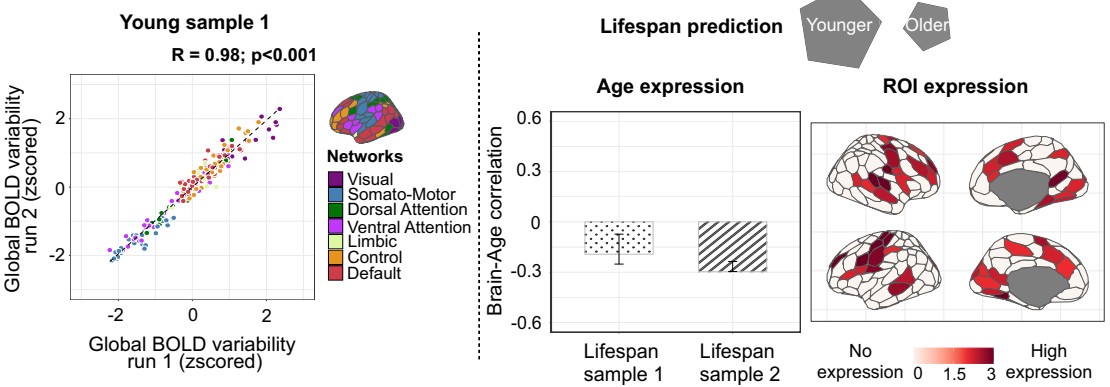

**C** **Validation of covSTATIS: high test-retest reliability & replicated age effects on global variability**

eigenvalue decomposition (Fig. 1B *step* 4) and obtained a multivariate connectivity space, wherein regions that showed similar connectivity values over time were closer together than regions with less similar connectivity values across windows (Fig. 1B *step* 5). covSTATIS next allowed us to back-project into this abstract multivariate Cartesian space, for every individual, each region's mean connectivity value over time across all windows (Fig. 1B *step* 6, blue dot) and around it, the

region's connectivity value for each window (Fig. 1B *step* 6, green dots). Our last step involved calculating, for each individual, the area of the hull around each regional mean connectivity over time (Fig. 1B *step* 7). A greater area of the hull indicates greater distance/spread in connectivity across windows and is therefore characteristic of regions with greater dynamic functional connectivity, that is, greater global BOLD signal variability.

**Fig. 1 | Local and global BOLD signal variability. A** Local BOLD variability was quantified via the root mean squared successive difference (rMSSD) on each normalized regional timeseries. **B** Global BOLD variability was quantified as dynamic functional connectivity obtained on windowed regional timeseries via covSTATIS. **C** On the left, we tested covSTATIS test-retest reliability (group-level results) on a sample of 145 healthy young adults who underwent two successive 10-min runs of multi-echo resting-state fMRI. Pearson's product-to-moment correlation was used (two-tailed test). On the right, we used partial least squares to test whether covSTATIS-derived global variability was reduced as a function of age, in two healthy adult lifespan samples. Error bars represent 95% confidence intervals from 1000 bootstrapped samples. Decreased global BOLD variability is reported in the aging literature using traditional methods. Note: all analyses and visualizations in this paper reduce the number of functional networks from 17 to 7, despite data being parcellated with the Schaefer 200-17 solution. To maintain spatial granularity while easing interpretation, we merged regions from different subnetworks into their principal network (e.g., visual central and visual peripheral into visual). Source data are provided as a Source data file. rMSSD root mean squared successive difference. covSTATIS covariance-structuring three-way statistical tables.

We next validated our method by assessing its consistency, test-retest reliability, and its ability to capture age effects typically reported in studies using traditional dynamic connectivity methods. We used resting-state fMRI data from one sample of 145 healthy young adults with two successive runs, and two independent cross-sectional healthy lifespan datasets (see "Methods" for details). For each sample and individual, we partitioned their regional timeseries data into windows of different length and derived covSTATIS' area of the hull scores (i.e., global BOLD signal variability) for each iteration. We found global BOLD variability to be highly consistent across window lengths, both at the individual level (two-way mixed effects ICC model > 0.82 for all samples; Supplementary Fig. S1) and at the group level ($0.92 <$ Spearman rho $< 0.99$, $p < 0.001$) in all samples. Global BOLD variability additionally showed high reliability at the group level ($r = 0.98$; $p < .001$; Fig. 1C) and moderate reliability at the individual level for all window lengths (average $r \sim 0.30$, $p < .001$, large Cohen's $d > 1.6$). In line with current literature demonstrating an age-induced dampening of the brain's dynamic range[66], here we used partial least squares (PLS)[67,68]—a multivariate method that assesses the covariance between two or more sets of variables—and showed that covSTATIS-derived area of the hull scores also decreased with age, particularly in regions that preferentially show age effects in the literature[69] (Fig. 1C; one significant latent variable (LV1) at $p = 0.003$ explaining 72% brain-age variance; lifespan sample 1 brain-age $r = -0.19$, lifespan sample 2 brain-age $r = -0.30$). Furthermore, age effects were stable across windows of different length, both in terms of their spatial location and directionality (LV1 at $p < 0.001$ explaining 58% brain-age variance; Supplementary Fig. S2). Together, these results highlight how covSTATIS is a valid and robust tool to estimate global BOLD signal variability.

**Bridging across fMRI datasets: reliability and topography of local and global BOLD signal variability**

We estimated local and global BOLD signal variability on two openly available resting-state fMRI datasets with diverse acquisition protocols. Given the heterogeneity of fMRI data used in the literature, it is imperative to quantify the dependency of local and global BOLD variability on the type of fMRI data used to extract them, to ensure generalizability. Here, we chose two datasets that differ in echo time and band acquisition. The number of echo times and bands influences the spatial and temporal resolution of fMRI, which in turn may impact the spatiotemporal properties of the underlying BOLD signal. Multi-echo acquisition allows for greater spatial coverage yet comes at the cost of slower acquisition time; multi-band acquisition allows instead for faster acquisition but is more susceptible to motion and scanner artifacts[70–73]. Throughout this paper, analyses were conducted on both samples in parallel. Young sample 1 refers to our multi-echo single-band fMRI dataset[74], and young sample 2 refers to our single-echo multi-band fMRI dataset[75] (see Fig. 2A and "Methods" for details about the samples).

Despite differences in the distribution of regional variability values across fMRI data type (Fig. 2A; rMSSD: young sample 1 mean(SD) = 23.73(9.85), IQR = 12.75; young sample 2 mean(SD) = 13.93(4.18), IQR = 5.16; covSTATIS: young sample 1 mean(SD) = 0.01(0.002), IQR = 0.002; young sample 2 mean(SD) = 0.02(0.003), IQR = 0.004; arbitrary units), local and global BOLD variability overall converged across samples (Fig. 2B, *top*). Greater regional and network reliability was found for global than local BOLD variability, as indicated by the greater number of regions and networks showing consistent mean values across fMRI samples (Fig. 2B, *middle* and *bottom*). For both local and global BOLD variability, reliability was lowest in sensorimotor (i.e., lower-order) areas and highest in heteromodal (i.e., higher-order), particularly default network, regions.

Building on these findings, we next described the spatial topography of local and global BOLD variability by fMRI data type. Understanding the spatiotemporal complexity of BOLD signal variability is necessary to build accurate computational and empirical models of brain function. Computational models typically treat macroscale signal variability as spatially homogenous to maximize mathematical tractability. Empirical fMRI studies mostly focus on the behavioral and clinical applicability of these measures. Furthermore, the known heterogeneity in spatial coverage across fMRI acquisition protocols is oftentimes overlooked.

For both fMRI data types, local and global BOLD variability showed a heterogeneous topography across brain regions (Figs. 3 and 4). Local BOLD variability presented greater topographical divergence across regions and networks than global BOLD variability, and its topography varied the most by fMRI data type (Fig. S3). Regional rank orders of local and global BOLD signal variability highlight the greater correspondence in higher-order cortices across fMRI data types (Fig. S4).

To contextualize these observed topographies, we mapped group-level regional profiles of local and global BOLD variability onto regional multi-scale maps of neocortical brain organization. Such maps were derived from open-source ex vivo cytoarchitectural[50], in vivo microstructural[51], ex vivo transcriptional (molecular)[52,53] and static functional connectivity MRI data[54]. Both local and global BOLD variability were found to be organized along multiscale gradients: both measures significantly mapped onto more than one neurobiological system (Fig. 5). This multiscale mapping was consistent across fMRI data type. Local BOLD variability was situated along an anterior-posterior gradient that was maximally associated with laminar and cellular spatial organization ($r = 0.52$, $p_{10\,k\,spin} < 0.001$ and $r = 0.62$, $p_{10\,k\,spin} < 0.001$). Global BOLD variability instead evolved along a unimodal-transmodal gradient that preferentially related to underlying regional microstructure and static functional connectivity ($r = -0.32$, $p_{10\,k\,spin} = 0.03$ and $r = 0.47$, $p_{10\,k\,spin} < 0.001$). Remarkably, these two distinct axes of local and global BOLD variability provide direct replication of previous work showing how intrinsic fMRI dynamics, estimated as the optimum combination of more than 6000 temporal BOLD timeseries features, collectively evolve along these two axes of neocortical brain organization[32].

**Bridging across spatial scales: local and global BOLD signal variability sit at the intersection of multiscale neocortical organization**

We next sought to comprehensively interrogate the multiscale properties of local and global BOLD signal variability. If variability is an emergent, crucial feature of complex systems, empirical fMRI

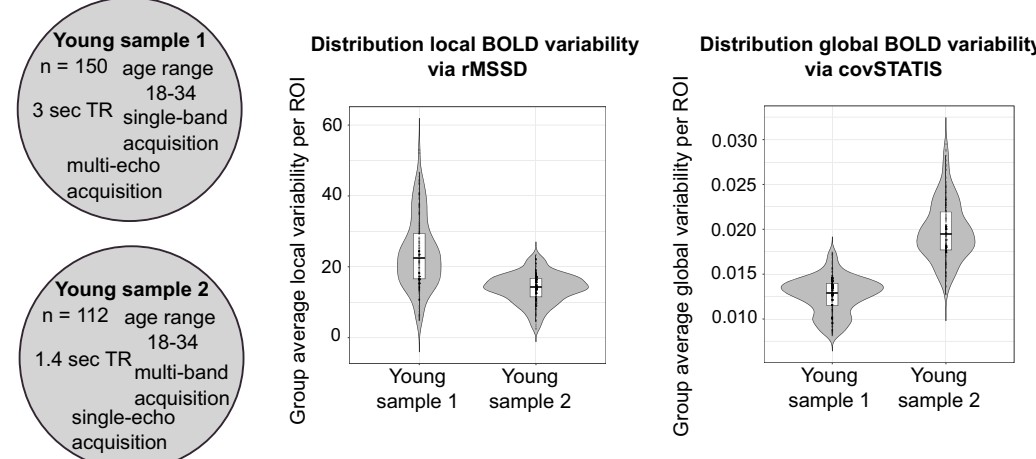

**A  Heterogeneity of fMRI data included in the study & distribution of measures of interest**

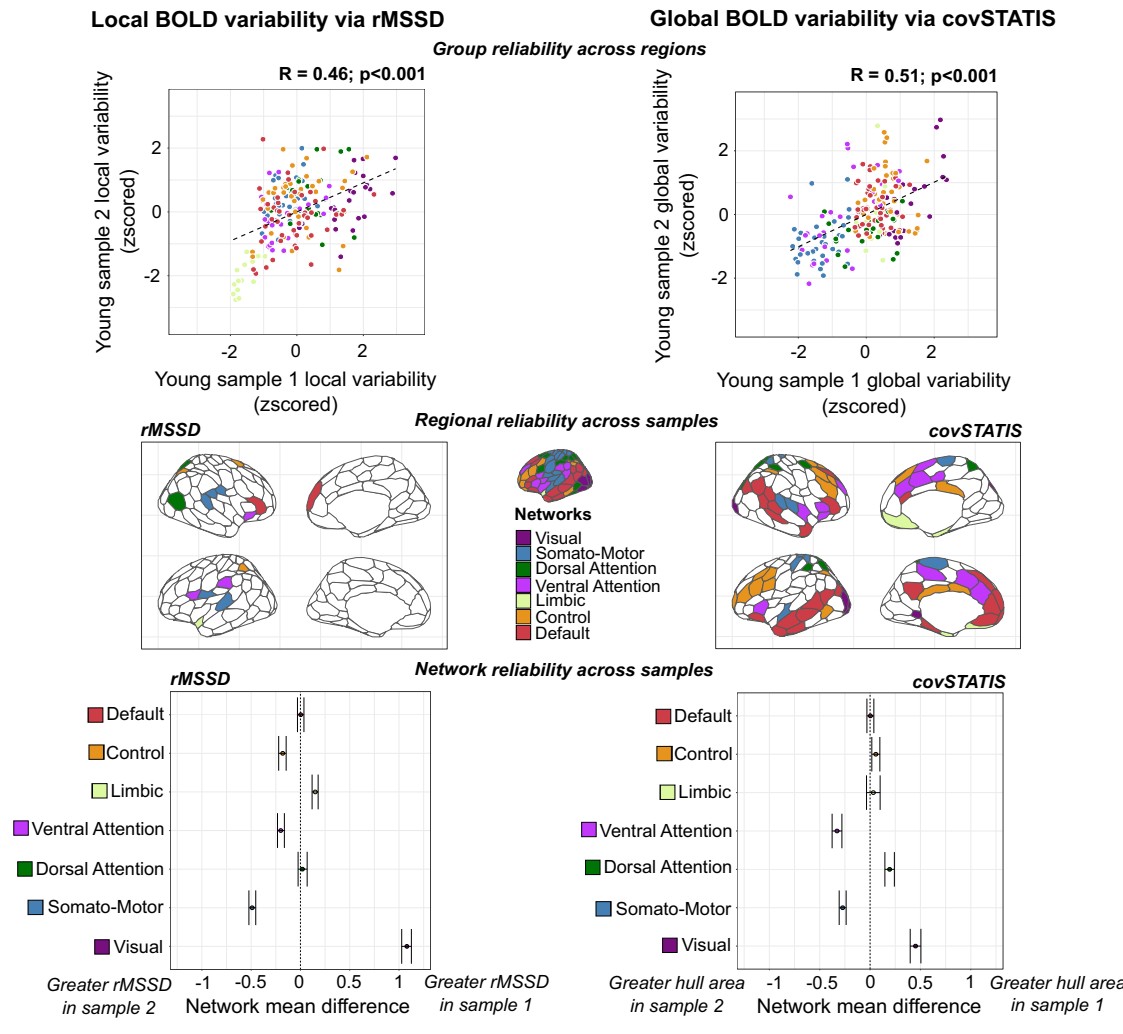

**B  Inter-sample reliability of local BOLD variability and global BOLD variability**

measures of brain variability should recapitulate core aspects of multiscale brain organization. Building on this inference, we hypothesized that local and global BOLD variability would sit at the intersection of multiscale neocortical organization (Fig. 6, *top left*). To test this hypothesis, we investigated whole-brain relationships between our measures of local and global BOLD variability and open-source micro-, meso-, and macro-scale neurobiological variables, for each fMRI data

type separately. Microscale neurobiological metrics included the first gradient of regional cytoarchitectural differentiation from ex vivo BigBrain histological data[50], and the first gradient of microstructural differentiation from in vivo quantitative T1 imaging from the MICA-MICs dataset[51]. Mesoscale neurobiological metrics were calculated as composite scores on PET-derived whole-brain neuroreceptor density maps available through Neuromaps[47]. We derived composite scores, as

**Fig. 2 | Distribution of local and global BOLD signal variability and their reliability across two different fMRI data types. A** On the left: description of the two fMRI samples used in this study. On the right: violin plots show the distribution of regional measures of local and global BOLD variability ($n = 200$ regions/observations) obtained on group-level data within each sample ($n = 150$ young sample 1, $n = 112$ young sample 2). More specifically, for each boxplot: the center line indicates the median value, the lower and upper bounds correspond to the 25th and 75th percentile, the whiskers represent the range of values within 1.5× the interquartile range, the minima and maxima delineate the min/max regional values within 1.5× the interquartile range, and individual points are the 200 cortical regions. **B** *Top*: inter-sample reliability of local (left) and global (right) BOLD variability across brain regions, estimated on group-level measures (Pearson's product-to-moment correlations; exact $p = 1.298e\text{-}11$ (left plot), $p = 1.725e\text{-}14$ (right plot)). *Middle*: regional reliability of local (left) and global (right) BOLD variability across samples. Independent two-sample Welch's t-tests (two-sided) were

computed for each brain region ($n = 200$ tests per metric). Each region contributed one observation per subject ($n = 150$ young sample 1, $n = 112$ young sample 2) to assess mean regional differences in local (left) and global BOLD variability (right) across fMRI data types. Colored regions show reliable effects ($p > 0.05$), that is regions with no significant differences between fMRI data types. Bottom: network reliability of local (left) and global (right) BOLD variability across samples. For each network, all region-level values were treated as observations and pooled together within sample (e.g., visual network: 24 regions × 150 subjects for young sample 1, 24 regions × 112 subjects for young sample 2). Two-sample Welch's t-tests (two-sided) were calculated ($n = 7$ tests per metric) to assess mean network differences in local (left) and global BOLD variability (right) across fMRI data types. Points show the network-level mean differences in local and global BOLD variability between fMRI data types (i.e., samples). Error bars represent 95% confidence intervals of these network-level mean differences. Networks crossing 0 show reliable effects ($p > 0.05$). Source data are provided as a Source data file. TR repetition time.

---

opposed to individual neuroreceptor maps, given the heterogeneity of the molecular and chemical composition of each brain region[76]. Macroscale neurobiological metrics comprised: PET-derived whole-brain maps of oxygen metabolism, glucose metabolism, cerebral blood flow and cerebral blood volume; and large-scale gradients of brain organization, including fMRI-derived sensory-association axis, the principal gradient of fMRI static functional connectivity and MEG-derived intrinsic timescale, all downloaded from Neuromaps[47]. Based on recent work on the role of BOLD temporal autocorrelation properties in recapitulating macroscale brain organization[77], for each of our fMRI samples, we additionally extracted group-level spatial maps of regional temporal autocorrelation scores from the BOLD signal. Such scores were derived as the product-to-moment correlation between successive (lag-1) and alternate (lag-2) timepoints of each regional timeseries tailored to each sample's TR. Lastly, to expand on recent evidence on the link between arrhythmic oscillatory brain activity and brain stochasticity[78], we leveraged open-source MEG data[79,80] and parametrized neurophysiological spectra[81], to spatially characterize whole-brain cortical arrhythmic brain activity and obtain a group-level spatial map. Each one of these group-level multiscale maps was separately correlated with group-level local and global BOLD variability maps from each of our fMRI samples, across regions. Statistical significance was assessed via 10,000 Hungarian spins on the regional parcellation of our local and global BOLD variability maps.

Overall, across fMRI data types, local and global BOLD variability were both associated with several neurobiological measures belonging to each spatial scale (Fig. 6, *middle* and *right*). At the microscale, greater local BOLD variability was associated with greater laminar differentiation ($r = 0.52$, $p_{10\text{ k spin}} = 0.007$ young sample 1). Specifically, greater rMSSD was present in regions showing heightened differentiation in cell size and density and clearer cortical layer separation. Motivated by previous work on the cytoarchitectural properties of static BOLD signal measures[33], we next derived cortical thickness for supragranular (mean across layers I–III), granular (layer IV) and infragranular layers (mean across layers V–VI) from BigBrain data[50], and related their whole-brain spatial distribution to our local BOLD variability spatial maps. We predicted local BOLD variability to be associated with the expression of granular layer IV in particular. Layer IV receives feedforward thalamo-cortical inputs[82,83]. If regions with a prominent layer IV need to orchestrate incoming feedforward projections, then such regions may exhibit greater variability in their functional activity. Theories about the thalamo-cortical pathway have proposed high local BOLD variability along these connections[84]. Additionally, layer IV is thickest in visual areas and absent in motor regions, precisely where we observed the highest and lowest levels of local BOLD variability. In line with our predictions, we found that greater local BOLD variability was significantly associated with stronger layer IV expression ($r = 0.39$, $p_{10\text{ k spin}} = 0.01$ young sample 1;

Fig. S5). At the mesoscale, greater local BOLD variability was related to greater ionotropic receptor density ($r = 0.39$, $p_{10\text{ k spin}} = 0.002$ young sample 1), reduced metabotropic receptor density ($r = -0.23$, $p_{10\text{ k spin}} = 0.008$ young sample 2), decreased receptor diversity ($r = -0.25$, $p_{10\text{ k spin}} = 0.002$ young sample 1), and decreased excitation/inhibition (E/I) ratio ($r = -0.47$, $p_{10\text{ k spin}} < 0.001$ young sample 1; $r = -0.31$, $p = 0.04$ young sample 2). At the macroscale, heightened local BOLD variability was associated with increased oxygen ($r = 0.41$, $p_{10\text{ k spin}} = 0.01$ young sample 1; $r = 0.54$, $p_{10\text{ k spin}} < 0.001$ young sample 2) and glucose metabolism ($r = 0.62$, $p_{10\text{ k spin}} < 0.001$ young sample 2), along with greater cerebral blood flow ($r = 0.50$, $p_{10\text{ k spin}} < 0.001$ young sample 2) and volume ($r = 0.41$, $p_{10\text{ k spin}} = 0.009$ young sample 1).

At the microscale, greater global BOLD variability covaried with thinner infragranular layers ($r = -0.45$, $p_{10\text{ k spin}} = 0.004$ young sample 1; Fig. S5). Greater global BOLD variability was also related to overall decreased microstructural differentiation ($r = -0.32$, $p_{10\text{ k spin}} = 0.03$ young sample 2). At the mesoscale, greater global BOLD variability was associated with increased metabotropic receptor density ($r = 0.33$, $p_{10\text{ k spin}} < 0.001$ young sample 1; $r = 0.26$, $p_{10\text{ k spin}} = 0.003$ young sample 2). At the macroscale, it was positively related to the principal gradient of fMRI static functional connectivity ($r = 0.47$, $p_{10\text{ k spin}} < 0.001$ young sample 2) and the sensory-association axis ($r = 0.35$, $p_{10\text{ k spin}} = 0.01$ young sample 2).

To quantify the robustness of these multiscale correlations across fMRI data type, we next quantified the degree of overlap in the neurobiological correlates of local and global BOLD variability across our two fMRI samples. We computed rank correlations on the Fisher-z transformed correlation vectors characterizing the relationships between local and global BOLD variability and multiscale variables for each fMRI sample. Both local and global BOLD variability exhibited strong overlap in their associations with multiscale neurobiological factors across fMRI data types (Fig. 6, *bottom*).

To quantify the central role of local and global BOLD variability in multiscale brain organization, we next ran cartographic analyses on our two sample-specific correlation matrices. Cartographic analyses are commonly used to derive graph theory metrics from brain networks[35]. We first assigned local and global BOLD variability, micro-, meso- and macro-scale measures to four different communities. We then took the absolute value of the reported correlations and calculated, for each fMRI sample, the participation coefficient of local and global BOLD variability[39,85]. Participation coefficient scores allowed us to determine how evenly distributed across spatial scales were the correlations of local and global BOLD variability, for each fMRI data type. Scores closer to 1 indicate greater multiscale participation[86]. We found high participation scores for both metrics and fMRI samples (young sample 1: local = 0.65, global = 0.71; young sample 2: local = 0.49, global = 0.59).

# Functional topography of local BOLD variability via rMSSD

## Young sample 1

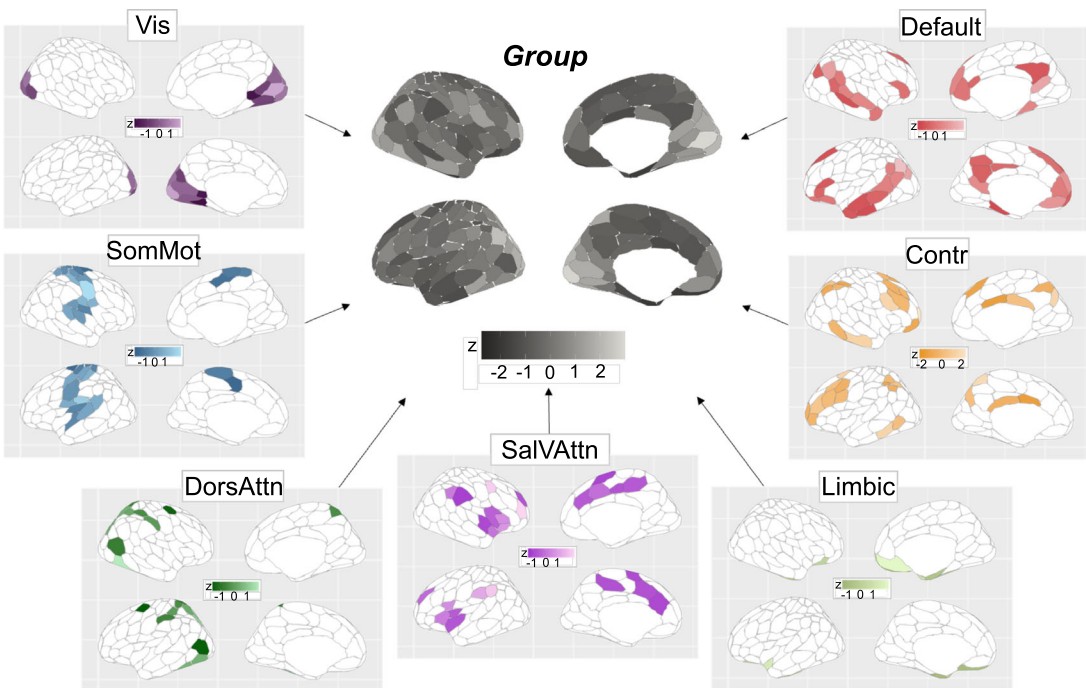

## Young sample 2

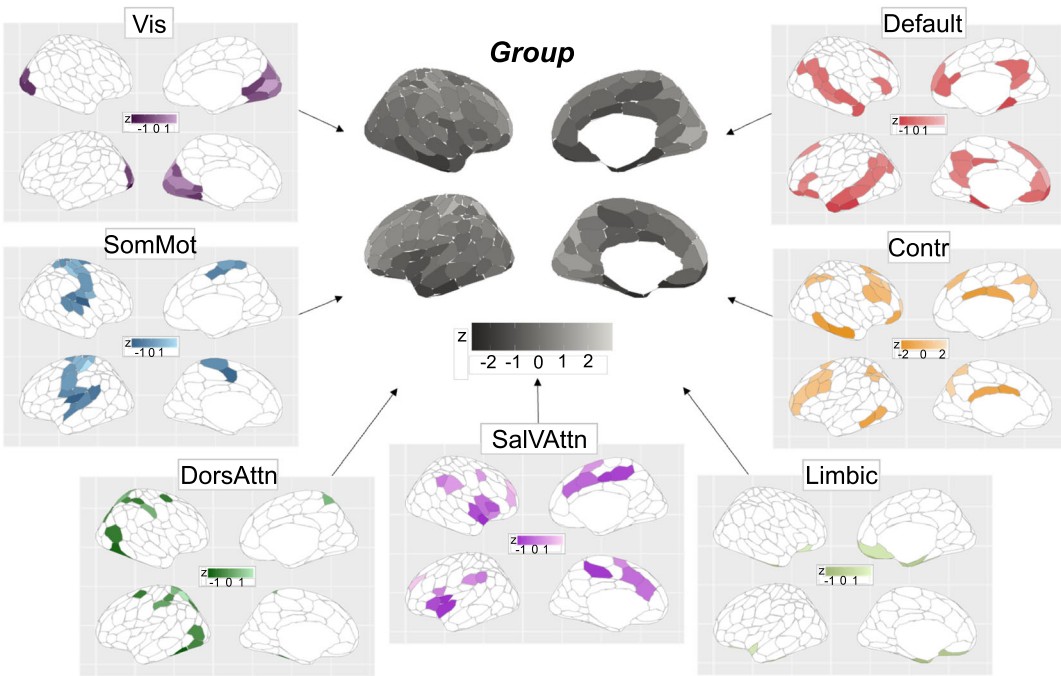

**Fig. 3 | Topographical characterization of local BOLD signal variability per fMRI data type.** Regional topography of local BOLD variability. Group-level spatial maps were obtained averaging BOLD variability maps across individuals within each fMRI sample. Central, grey-scale maps are the group averages. To ease interpretation, we show around the central maps the 7 canonical network-level group maps. To appreciate between-network differences in BOLD variability, we let scaling values differ between networks. Note that rMSSD values were z-scored for easier comparison across datasets as units are arbitrary. Source data are provided as a Source data file. Vis visual network. SomMot somato-motor network. DorsAttn dorsal-attention network. SalVAttn salience ventral-attention. Contr control.

## Functional topography of global BOLD variability via covSTATIS

### Young sample 1

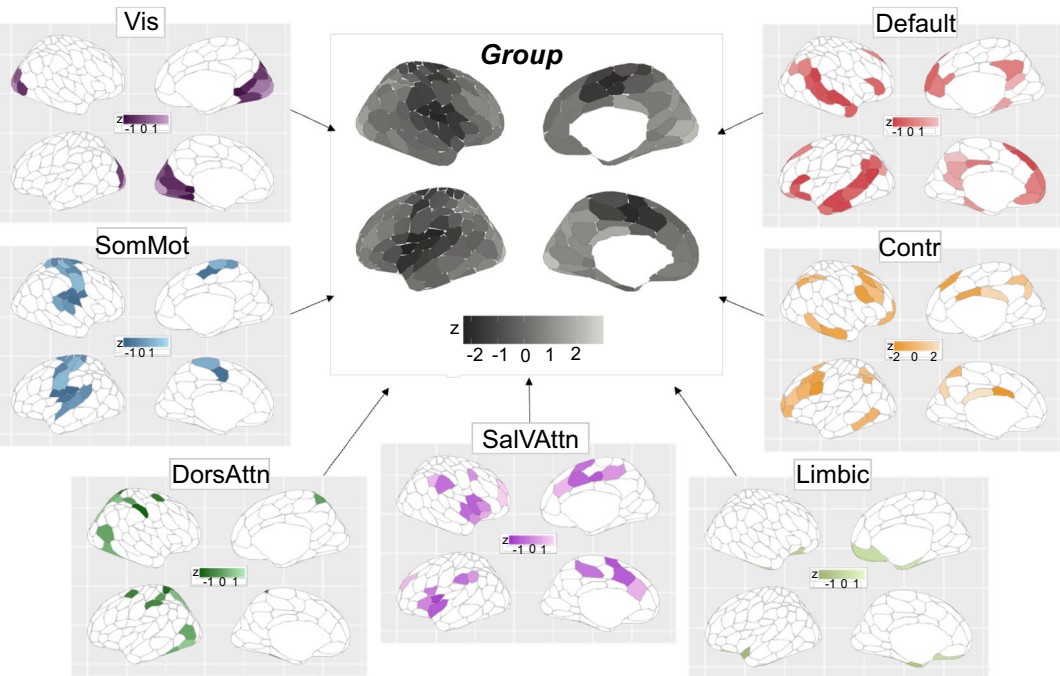

### Young sample 2

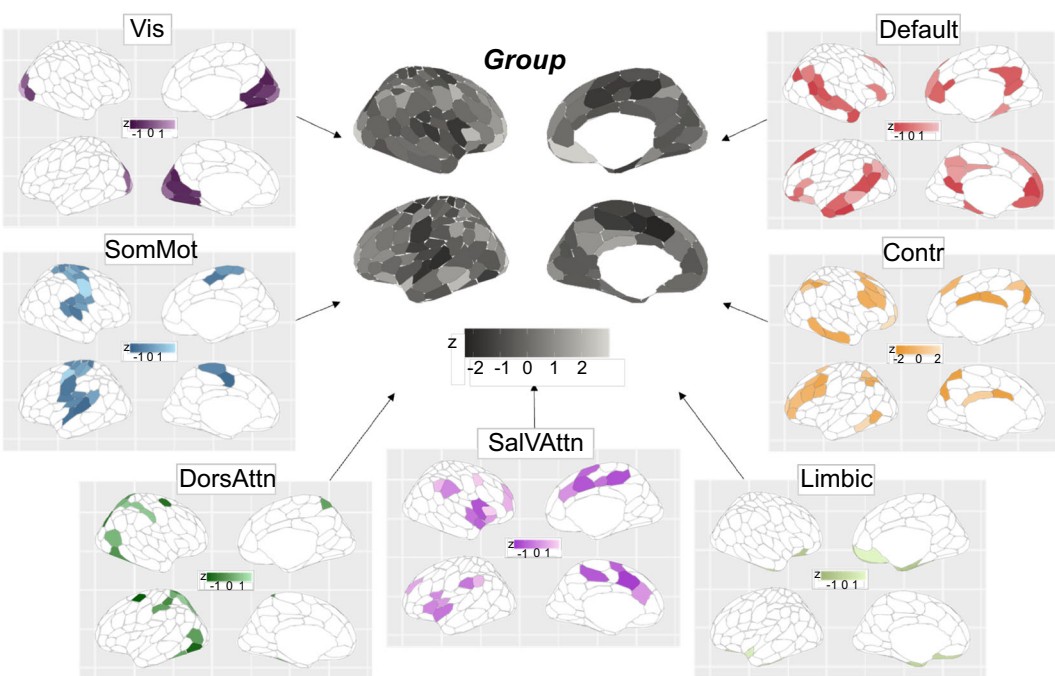

**Fig. 4 | Topographical characterization of global BOLD signal variability per fMRI data type.** Regional topography of global BOLD variability. Similarly to local BOLD variability, we show group-level spatial maps of dynamic functional connectivity both for the whole brain and the canonical 7 networks (z-scored values). Source data are provided as a Source data file. Vis visual network. SomMot somato-motor network. DorsAttn dorsal-attention network. SalVAttn salience ventral-attention. Contr control.

To further validate the multiscale nature of local and global BOLD variability, we used dominance analysis (DA) (see "Methods" for details) to build a predictive model for each fMRI dataset, where we estimated the unique contribution of each neurobiological measure in predicting local and global BOLD variability[18,87]. In line with our correlational results, we found that both local and global variability were predicted by a combination of neurobiological variables within each sample (Fig. 7).

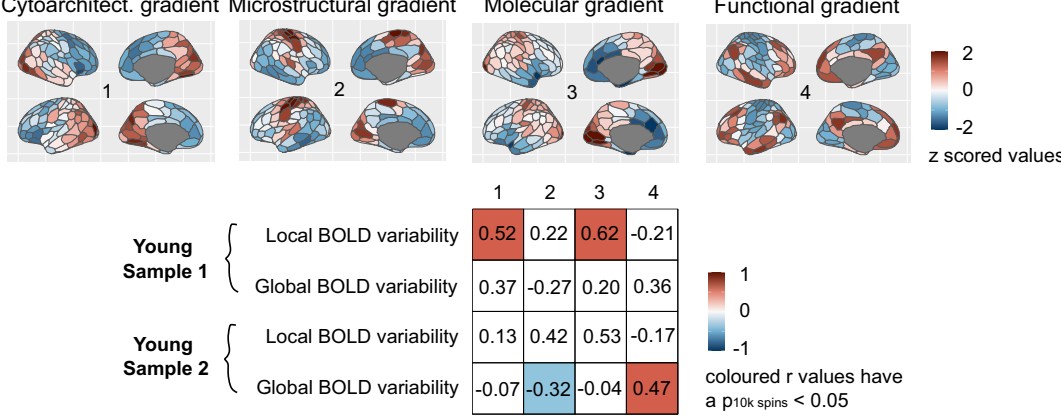

**Fig. 5 | Multiscale gradients of local and global BOLD variability.** We correlated our group local global BOLD variability maps with open-source ex vivo cytoarchitectural, in vivo microstructural, transcriptional and static functional connectivity maps, for each fMRI sample separately. Pearson's product-to-moment correlations were used, and significance was assessed via permuting 10,000 times the regional labels of our local and global variability spatial maps (Hungarian spins; two-sided spin permutation test). The table shows the resulting correlation values split by metric and sample. Colored boxes indicate significant correlations ($p_{10\,k\,spin} < 0.05$). Source data are provided as a Source data file.

As a final step, we repeated all analyses above, including the second gradient of functional connectivity organization[54]. Broad patterns of association were largely maintained across the two gradients, yet the principal gradient showed more reliable inter-sample results (Fig. S6).

**Bridging across temporal scales: local BOLD signal variability is anchored in electrophysiological processes**

To further understand the temporal and neuronal properties of BOLD signal variability, we turned to electrophysiological data and used a combination of open-source MEG source-modeled, broadband (1–150 Hz) resting-state data[79,80] and simulations of naturalistic electrophysiological timeseries, to derive and characterize local measures of brain variability. We decided to compute only measures of local variability and not of global variability, in light of the known methodological variance of functional connectivity derivatives in the MEG literature[88].

We derived two measures of local brain signal variability on MEG regional timeseries, based on how variability is independently characterized in fMRI and MEG: (1) moment-to-moment signal intensity changes via rMSSD (MEG signal variability), and (2) 1/f exponent, that is, the slope of the MEG power spectrum, shown to capture background arrhythmic activity[89,90]. We first looked at the spatial topography of both measures. For the former, we observed the highest variability levels in somato-motor areas, lowest values in visual and dorsal attention regions, and mid-levels in higher-order cortices (Fig. 8A; network maps in Fig. S3). For the latter, we replicated previous reports showing steeper slopes (i.e., greater 1/f exponent values) in posterior cortical regions[89] (Fig. 8B, *left*).

Next, for each individual, we obtained whole-brain measures of local MEG variability for both rMSSD and 1/f exponent measures by averaging across brain regions. We then related the two variables across individuals and found a strong negative association between them ($r = -0.6$; $p < .001$, Fig. 8B *middle*): individuals with a flatter 1/f exponent showed heightened levels of local rMSSD-derived MEG variability. To expand on these findings, we built a hierarchical linear model where we tested for rMSSD-1/f exponent relationships while accounting for regional heterogeneities in both measures (Fig. S7). The model showed regional diversity in the association between rMSSD and the 1/f exponent (Fig. S7). To further get a mechanistic

understanding of this relationship, we simulated naturalistic electrophysiological timeseries using the NeuroDSP toolbox[91] (see "Methods" for details). We manipulated the steepness of the 1/f exponent at various parameters ranging from −0.7 to −1.5 in steps of −0.1. 10 simulations were run per step, and local rMSSD-derived MEG variability was calculated for each manipulation. We found that flatter 1/f exponents were strongly predictive of greater local MEG variability ($R^2 = 0.97$; $\beta$(SE) = −9.86(0.02); CI [−9.51; −9.10]; $p < 0.001$; Fig. 8B *right panel*). These results highlight the arrhythmic nature of electrophysiological signal variability.

As a final step, we tested cross-modal relationships between local fMRI and MEG signal variability. We report both cross-sample and within-sample associations (i.e., fMRI-MEG data obtained on the same individuals from the MEG sample with available fMRI data). At the group level, we found a consistent, negative association across and within sample between fMRI and MEG variability (Fig. 8C). A similar negative association was also observed at the individual level (average Fisher z-to-r = −0.14; $t$(102) = −11.91, $p < 0.001$, 95% C.I. = [−0.17, −0.12], Cohen's $d = -1.17$). Together, these findings reinforce the biological multimodal nature of local brain signal variability.

## Discussion

Variability is a fundamental functional property of complex systems, as it determines system organization and behavior. The human brain is a complex stochastic system[16], hence integrating functional variability of brain signals is essential for the modeling of human brain function. In this study, we investigated the spontaneous local and global variability of the fMRI BOLD signal. We comprehensively characterized the statistical properties, multiscale topographies and neurobiological components of local and global BOLD signal variability, respecting the complexity of the fMRI BOLD signal. By bridging across fMRI data types and spatial and temporal scales, we showed that macroscale measures of local and global BOLD signal variability are integral aspects of brain function. Local and global BOLD signal variability are not merely sources of biologically irrelevant noise. They are reduced with age in line with most prior investigations[19–21,25–27,29,92–94], though some studies have reported opposite effects[95,96]. We have shown how measures of BOLD signal variability have a spatially heterogeneous topography, encapsulate micro-, meso- and macro-scale neurobiological phenomena, and are related to underlying electrophysiological

## Local and global BOLD variability map onto multiscale neurobiological metrics

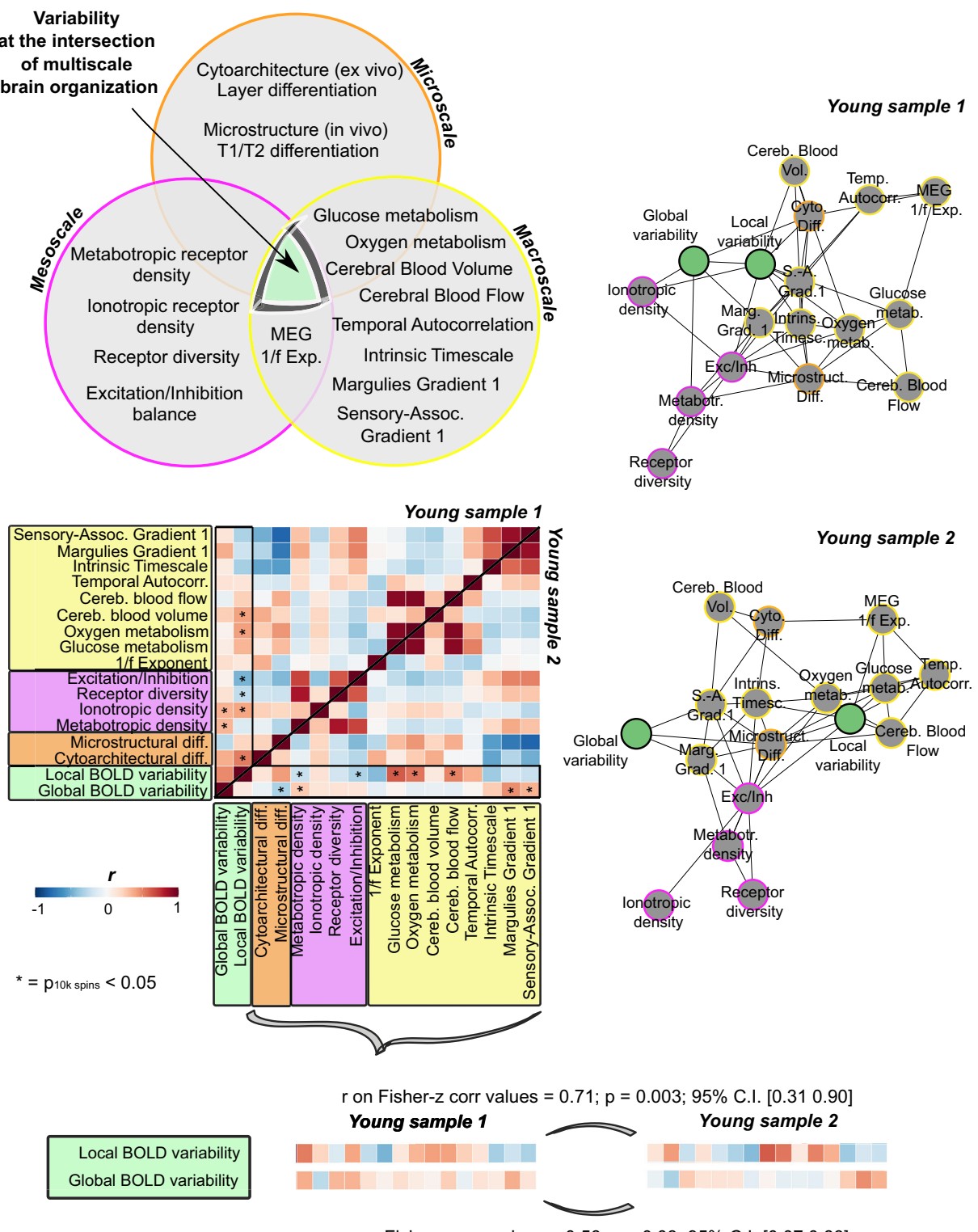

**Fig. 6 | Multiscale neurobiological correlates of local and global BOLD signal variability.** *Top left:* Euler diagram representing our hypothesis of the central role of local and global BOLD variability in multiscale brain organization. Each circle represents a spatial scale and includes all variables used in our analyses. *Middle left:* correlation matrices for each fMRI sample (upper triangle: young sample 1; lower triangle: young sample 2). Pearson's product-to-moment correlation was used. Asterisks indicate correlations that survived significance testing (10,000 Hungarian

spins of Schaefer's regional labels; two-sided spin permutation test). *Right:* spring embedding plots represent correlations with an absolute value above 0.3, for each fMRI sample. Note that link length reflects correlation magnitude. *Bottom:* Pearson's product-to-moment correlations (two-sided) between the multiscale correlates of local and global BOLD variability across fMRI samples. Multiscale correlates were first Fisher-z transformed before being related across fMRI samples. Source data are provided as a Source data file.

# Predicting local and global BOLD variability from multiscale neurobiological metrics

Local BOLD variability via rMSSD

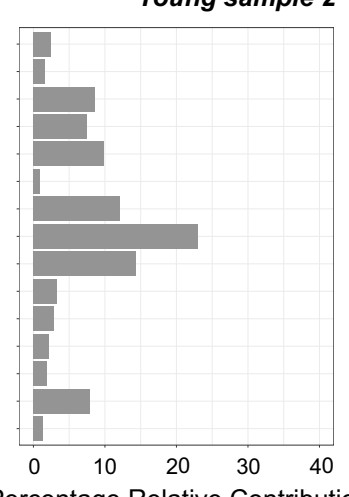

Global BOLD variability via covSTATIS

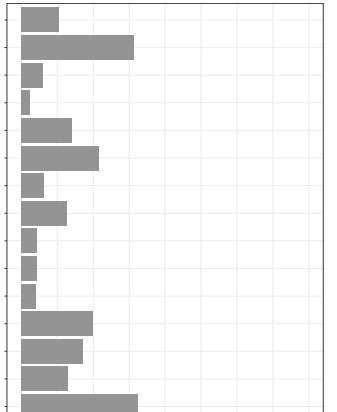
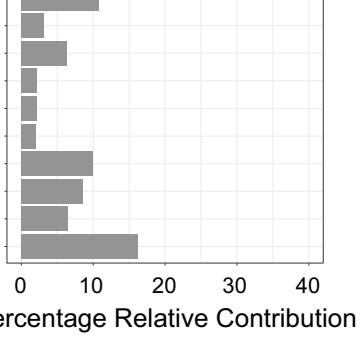
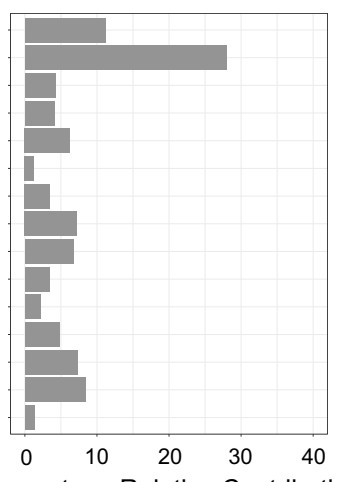

**Fig. 7 | Multiscale neurobiological predictors of local and global BOLD signal variability.** Dominance analysis results per metric and fMRI sample. Results indicate the unique contribution of each neurobiological variable in predicting local and global BOLD variability and recapitulate our correlational results. Source data are provided as a Source data file.

neuronal activity. Our findings motivate cognitive network neuroscience research to re-evaluate local and global BOLD variability as structured, multifactorial, heterogeneous properties of human brain organization, and offer a comprehensive lens on brain stochasticity across spatial and temporal scales.

Brain signal variability has been increasingly recognized as a fundamental feature of neural functioning. At the cellular and molecular level, studies have shown how neural variability serves as preparatory activity for future stimulus processing. It allows for the recapitulation of previous sensory experience[97–99], predicts ongoing behavioral and cognitive states[100–104], facilitates synaptic formation[100,105–107], and drives signal propagation[108]. At the macroscale level in fMRI, accumulating evidence has highlighted BOLD signal

variability as a key feature of optimal behavioral functioning and a marker of neurocognitive aging and clinical status[20,21,25–27,29,92–95,109–113]. Our work builds on these multiple lines of evidence and further expands the relevance of brain signal variability from a multiscale neurobiological standpoint.

Local and global BOLD variability are statistical approximations of biological processes unfolding over time. It is thus important to consider how their statistical behavior may explain the effects we observe empirically. Local BOLD variability is computed at the level of single regional BOLD timeseries, whereas global BOLD variability involves the interpolation of pairs of local BOLD timeseries. Consequently, local BOLD variability is closer to the data from which it is derived than global BOLD variability, explaining its greater dependency on fMRI

## A Functional topography of local MEG variability via rMSSD

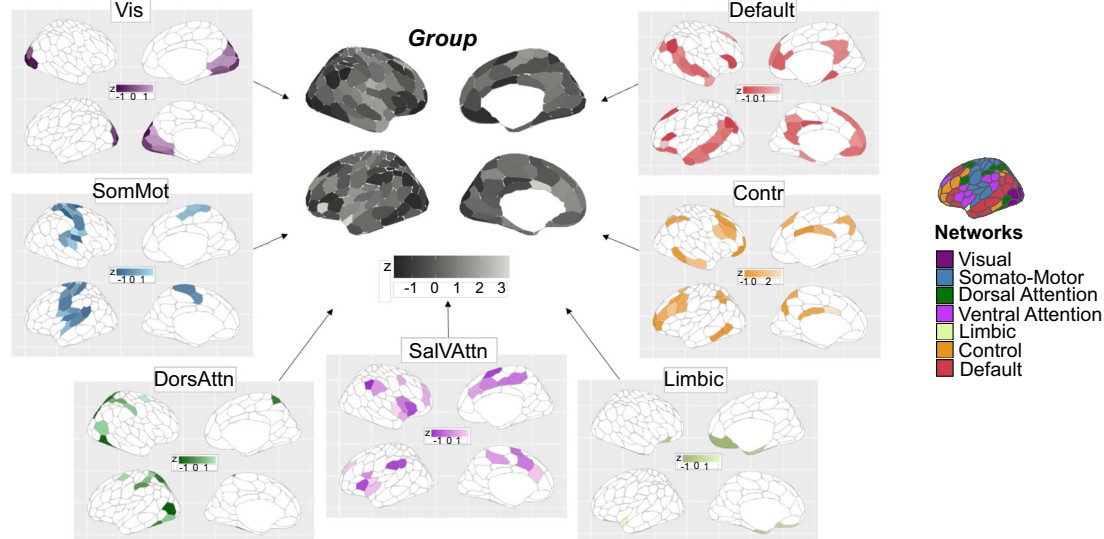

## B Functional topography of 1/f Exponent & relationship with local MEG variability

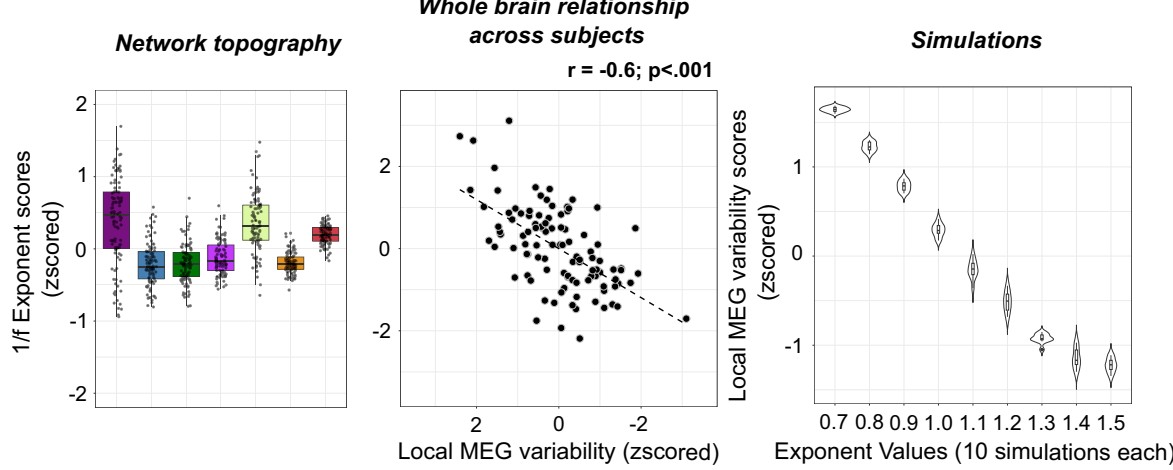

## C Local fMRI BOLD variability relates to local MEG variability

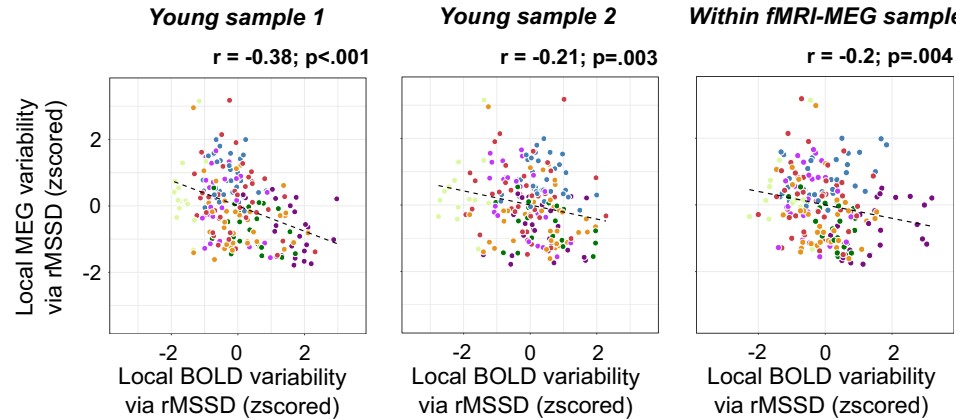

data type. Levels of statistical approximation are intuitively linked to levels of brain organization: lower levels of statistical approximation allow for more specific local neural representations taking place in unimodal lower-order regions, while higher levels of statistical approximation reflect lower-dimensional global neural representations residing in heteromodal higher-order areas. Building on Mesulam's visionary work[114], this statistical explanation elucidates why, in this study, lower-order regions exhibited lower reliability of variability measures than higher-order areas across fMRI samples. Across primate species, lower-level areas have spatially smaller

**Fig. 8 | Contextualizing local brain variability across neuroimaging modalities.** Local electrophysiological variability was estimated as MEG signal variability via rMSSD, and as the 1/f exponent in the MEG power spectrum. **A** Functional topography of local rMSSD-derived MEG variability. Central, grey-scale map represents the group average map. Around it are the 7 canonical network-level group maps. Note that rMSSD values were z-scored, as units are arbitrary. **B** *Left*: functional topography of the 1/f exponent per network, across individuals. For each subject ($n = 104$), regional 1/f exponent values were normalized within subject and averaged within each network (7 values per subject). Boxplots thus show the between-subject distribution of network-mean 1/f exponent measures, with each point representing one subject. Boxplots are defined as follows: the center line indicates the median value, the lower and upper bounds correspond to the 25th and 75th percentile, the whiskers represent the range of values within 1.5× the interquartile range, and the minima and maxima delineate the min/max values within 1.5× the interquartile range. *Middle*: whole-brain relationship between local rMSSD-derived

MEG variability and 1/f exponent across individuals. Pearson's product-to-moment correlation was used (two-tailed test; exact $p = 2.362e\text{-}11$). *Right*: simulated naturalistic electrophysiological timeseries with systematically varied 1/f exponents. Timeseries were generated by varying the steepness of their 1/f exponent at various parameters from −1.5 to −0.7, shown as positive values in the graph, in steps of −0.1. 10 simulations were run per parameter ($n = 90$ total simulations). We calculated local rMSSD-derived MEG variability on each simulated timeseries. Violin plots show the distribution of rMSSD variability values across simulations for each exponent condition. **C** Cross-modal spatial relationships between local fMRI and MEG brain variability, between two independent samples (left and middle figures) and within the same sample using fMRI-MEG data (right figure). Pearson's product-to-moment correlation was used (two-tailed test; left plot's exact $p = 8.751e\text{-}08$). Source data are provided as a Source data file. Vis visual network. SomMot somatomotor network. DorsAttn dorsal-attention network. SalVAttn salience ventral-attention. Contr control.

receptive fields and resonate at faster timescales[115,116]. These two features allow sensory-motor cortices to tune to quickly changing sensory-motor stimuli. Specificity and speed of sensory perception and motor action require minimal number of computational steps, resulting in variable outputs prone to local changes. In contrast, association regions have spatially larger receptive fields and oscillate at longer timescales[115,116]. These two features allow the association cortex to respond and integrate information across regions, and thus drive feedback processes in the brain[117]. Information integration across space and time is a higher-order computational process which relies on multiple upstream units, resulting in stable outputs robust against local changes.

Here, we showed how the statistical dependencies of local and global BOLD variability meaningfully interact with the statistical principles that govern the brain's spatial organization: cortical areas devoted to local processing presented variability patterns that were more dependent on fMRI data type for both local and global variability measures than higher-order regions. These results call for more targeted empirical research on the complex interaction between local-global cortical processing, local-global BOLD signal variability and fMRI data type. While direct in-depth investigations on the commonalities and differences between low- and high-level cortical areas in the context of local and global BOLD variability are needed, here we provide a first general demonstration of the multiscale features of local and global BOLD signal variability, as independently assessed. Reliable local and global BOLD variability features are thus separately discussed here in turn.

We found local and global BOLD variability to be spatially heterogeneous multiscale processes that closely map onto local and global information processes in the brain. While some inconsistencies emerged across fMRI samples due to inter-sample differences in the topography of local and global BOLD signal variability, their multiscale properties were largely conserved across samples.

Local variability unfolded along an anterior-posterior gradient, whereas global variability followed a unimodal-heteromodal gradient[118]. These findings shape the design of future empirical studies and computational models on local brain variability. Local brain variability has often been treated as a nuisance, extrinsically-driven, artefactual process and simplified as a spatially homogenous error term in models of brain function. Accumulating evidence has, however, shown how local BOLD variability defines global network structure over space[18,44,45] and time[119]. Our study adds to this body of evidence by revealing that local BOLD variability is a structured, spatially heterogeneous process. Computational approaches to brain dynamics may thus benefit from introducing spatial heterogeneity in local measures of variability to more precisely approximate empirical observations. In doing so, empirical fMRI data will not only serve as a model validation tool but also as a model optimization tool. While

early population models assumed and enforced spatial homogeneity in local dynamics[120–123], recent years have progressively seen the integration of spatial heterogeneity—from receptor density, gene expression, myelination, cortico-thalamic functional gradient maps—in neural mass models, to more accurately capture key features of brain dynamics observed empirically, including covariance, dynamic functional connectivity, regional times-cales, and stimulus sensitivities[14,15,34,124–128]. It is in this context that we argue that our results will help shape existing and future neurobiologically-informed computational models, by further constraining them with a new spatially heterogenous axis of brain dynamics.

Local and global BOLD variability are macroscopic representations of stochastic processes unfolding at finer spatial and temporal scales. A multi-scale contextualization of these measures fundamentally elevates the validity and applicability of fMRI imaging and significantly advances our understanding of BOLD signal variability. This study does so by identifying the neurobiological underpinnings of local and global fMRI BOLD signal variability across spatial and temporal scales. We showed that local BOLD variability emerges from a mixture of local cellular, molecular, neurochemical and neurovascular factors. Expanding on existing evidence[93], local BOLD variability is thus not entirely driven by BOLD physiological processes, but is instead the macroscopic representation of low-level micro- and meso-scale feed-forward operations in the brain. We found associations between greater levels of local BOLD variability and greater cytoarchitectural differentiation, thicker layer IV expression, greater neuronal density, and higher neurogenesis. We observed greater local BOLD variability in granular cortical regions. Granular regions are involved in feedforward processing and are directly innervated by the core cells of the thalamus[117]. The distinct laminar structure of granular regions enables segregation and efficient processing of incoming sensory information[129], facilitated by heightened neuronal density and neurogenesis. Increased cellular diversity and higher number of processing units signify a greater range of inputs in granular regions. Increased local BOLD variability in granular areas reflects greater temporal diversity of their functional responses, and therefore a greater range of outputs. Put together, these results indicate that a greater range of outputs may arise from a greater range of inputs. The elevated dynamic range of responses observed in these sensory regions may facilitate the detection of diverse environmental stimuli, a physical process called stochastic facilitation[130,131]. Sensory regions are known to maintain high stimulus fidelity by keeping incoming information separated and coupled to its environmental sources[114,132]. The high moment-to-moment stochastic properties of sensory cortices may enable them to effectively parallel in their processing, and be sensitive to, environmental uncertainty.

Greater local BOLD variability was also related to greater ionotropic—primarily inhibitory- receptor density. Ionotropic receptors are

fast-enacting receptors whose effects induce immediate local changes[133]. Local BOLD variability is maximal in fast-oscillating sensory cortices. The mesoscopic features of ionotropic transmission may consequently give rise to, and shape, the topography and timescale of macroscale local variability observed in fMRI. Additionally, the inhibitory signature of local BOLD variability complements existing literature on the direct involvement of GABAergic and dopaminergic (D2 inhibitory) receptors and thalamo-cortical GABAergic projections, in orchestrating local brain variability[24,26,28]. It is important to consider that measures of macroscale neurotransmitter receptor density do not, however, provide information about the ultimate mesoscopic effects that receptors have on neurotransmission. If, for instance, an excitatory receptor is located on an interneuron, when stimulated, this receptor will have an inhibitory effect on its target. Pharmacological interventions coupled with multi-scale imaging techniques are therefore necessary to more precisely characterize the valence of mesoscale aspects of local brain variability.

Global BOLD variability emerged as the macroscopic representation of high-level micro and mesoscale feedback operations in the brain. Global BOLD variability was maximal in higher-order cortices and, as a result, it was associated with increased microstructural similarity, greater metabotropic receptor density and higher values on the unimodal-heteromodal static functional connectivity gradient. These multiscale properties of global BOLD variability mirror the multiscale features of heteromodal regions: greater similarity in myelin content, along with slower and long-lasting neurotransmission, may facilitate information integration across the brain. Association cortices receive diffuse projections from the matrix cells of the thalamus, and via long-range connections are involved in feedback modulatory processes in the brain[117]. To be able to coherently update incoming information from lower-level cortices, feedback processes require flexible inter-regional connections, ultimately explaining why global BOLD variability was highest in association areas. Despite the impact of fMRI acquisition sequences on local and global BOLD variability, the centrality of these measures in multiscale brain organization was reliable across fMRI datasets, further establishing their biological relevance. More research is needed to understand how these multiscale features relate to the brain's spatiotemporal architecture on an individual-by-individual basis, given that our analyses were conducted at the group level and therefore disregarded intra-sample sources of variance.

We concluded our study by bridging across temporal scales and neuroimaging modalities. Given fMRI's indirect estimation of neuronal activity, we used temporally-rich electrophysiological signals to interrogate the neuronal nature of local BOLD variability. Via empirical and simulated MEG data, we first found local signal variability to be driven by the slope of the signal power spectrum. Our findings add to an emerging body of work highlighting how the previously disregarded slope of the electrophysiological power spectrum is biologically and behaviorally relevant[89,134–136]. Greater signal variance was observed in flatter power spectra—that is, spectra wherein all frequencies, even the highest, were represented. Since flat power spectra are characteristic of white noise, these results additionally point towards the potential biological role of higher frequencies and white noise in shaping local brain signal variability.

Via cross-sample and within-sample comparisons, we observed a consistent negative association between fMRI-derived and MEG-derived local brain signal variability. These results expand on previous literature showing a spatial correspondence between fMRI and MEG signals using static functional connectivity[137–141]. Our findings further demonstrate that fMRI and MEG signals share meaningful information about human brain dynamics. Here, the negative direction of fMRI-MEG correlations may be explained by differences in the topographies of the two signals, as they originate from distinct physiological sources and have different frequency content[142–144]. The relatively modest magnitude of their correlation may be due to differences in study design between the two modalities: MEG data were acquired during rest, while fMRI data during movie watching[79]. While further work is needed to shed more light on the nature of these relationships, our work provides preliminary evidence in favor of the biological multimodal nature of local brain signal variability. Specifically, these findings urge the general neuroscience community to reconsider what is empirically deemed noise (i.e., slope of the electrophysiological power spectrum, local BOLD signal variability) as biologically meaningful properties of multimodal brain signals.

Finally, we acknowledge the wide array of brain signal variability and complexity metrics in the fMRI and MEG literatures[18,103,145], yet in the absence of a systematic framework for inter-metric comparisons, this study focused on two of the most popular fMRI signal variability metrics (i.e., rMSSD and dynamic functional connectivity). A logically structured grouping of variability measures is necessary to disambiguate the ontological confusion in the field and enable robust and interpretable inter-metric comparisons. Different variability metrics are derived from different mathematical formulations with inherently different assumptions: in other words, variability measures have different epistemologies, and therefore they lead to distinct ontologies[146]. In the presence of such framework, future research should test how the reported multiscale patterns of local and global brain signal variability may depend on the choice of the variability metric.

Overall, the value provided by our measures of local and global variability is their relatively straightforward implementation, their mathematical simplicity (i.e., rMSSD captures processes related to basic distributional properties of a timeseries), and user-independent nature. We expect our rMSSD findings to hold across other basic timeseries measures (i.e., standard deviation), and to relate to more complex measures given the slow, linear nature of fMRI (i.e., (f) ALFF[147]). We also expect our covSTATIS findings to be recapitulated by more standard sliding window approaches[64], given their relatively similar implementation.

Altogether, via an integrative multi-data, multi-scale, multi-modal approach, our work distills the rich spatiotemporal information present in the fMRI BOLD signal and conveys the empirical biological properties of local and global BOLD signal variability. By relating BOLD signal variability to neurobiological and neurophysiological processes, we ascertained the accessibility of the BOLD signal to brain stochasticity at various spatial and temporal scales. By establishing BOLD signal variability as a spatially heterogeneous, multifactorial, multimodal property of brain organization integral to healthy brain function, this study underscores the value of systems-level approaches to brain function[148]. Our work provides fertile soil for new theoretical and methodological work in cognitive network neuroscience by opening the door to neurobiologically grounded hypotheses on the role of local and global BOLD signal variability in cognition and behavior. Moreover, our findings offer empirically grounded targets for generative models of brain signal dynamics in health and disease.

## Methods
### Neuroimaging datasets
All main analyses were carried out on two open-source resting-state fMRI datasets: young samples 1 and 2. To validate covSTATIS as a global BOLD signal variability estimation method, we leveraged two adult lifespan datasets: lifespan samples 1 and 2. To evaluate the temporal and neuronal properties of BOLD variability and to bridge across neuroimaging modalities, we used an open-source resting-state MEG dataset. Below, we briefly describe each neuroimaging dataset.

### fMRI young sample 1
A total of 150 healthy young individuals ages 18–34 ($M_{age}$ = 22 years, $SD_{age}$ = 3 years, 55% F) from the Neurocognitive Aging Dataset[74] were included in this study. Gender information was not available, and all

analyses were carried out on the whole sample (both female and male data together). All individuals in the study were right-handed, healthy, with no evidence of neurological, psychiatric, or other medical conditions impacting neural functioning[69]. Participants underwent two multi-echo resting-state fMRI scans of 10-min duration each within the same session. Analyses conducted in this paper were performed on the first run of data. The second run was used to assess test-retest reliability of global BOLD signal variability on 145 individuals, since 5 participants out of the total sample did not have a second run of data ($M_{age}$ = 22 years, $SD_{age}$ = 3 years, 55% $F$). Data were collected at the Cornell Magnetic Resonance Imaging Facility, at Cornell University (New York, US). All participants provided written informed consent and were compensated for their time. Research protocols were approved by the Cornell University Institutional Review Board.

Resting-state fMRI data were acquired on a 3T GE discovery MR750 using a multi-echo EPI sequence with online reconstruction (TR = 3000 ms; $TE_1$ = 13.7 ms, $TE_2$ = 30 ms, $TE_3$ = 47 ms; 83° flip angle; matrix size = 72 × 72; FOV = 210 mm; 46 axial slices; 3 mm isotropic voxels; 204 volumes) with 2.5× acceleration and sensitivity encoding. Participants were instructed to keep their eyes open during the two scans. Multi-echo resting-state fMRI data were preprocessed and denoised using multi-echo independent component analysis (ME-ICA v3.2; https://github.com/ME-ICA/me-ica)[70,71], as in our previous publication[69]. Briefly, functional images were minimally preprocessed (i.e., removal of the first 4 volumes, deobliquing, motion correction, anatomical-functional co-registration, spatial alignment of volumes across echoes-TEs). T2* maps were then used for anatomical-functional co-registration. Volumes were optimally combined across TEs and denoised. Image quality was lastly carried out on the denoised time series to check for co-registration and residual noise (framewise displacement > 0.50 mm, DVARS > 1, tSNR < 50, or fewer than 10 total included BOLD components). Functional images were parcellated using the group prior individual parcellation (GPIP)[69,149], a participant-specific parcellation approach initialized on the Schaefer 200-17 network solution[56]. We selected GPIP as our parcellation method of choice to ensure high topographical precision of fMRI BOLD, and for consistency with previous work from our group using the same data[69]. Unlike standard parcellations, GPIP accounts for within-subject variation in parcel boundaries, offering substantially improved sensitivity to individual differences in brain activity and behavior[149–152]. Specifically, GPIP consists of an iterative Bayesian process: it starts from the user-provided reference group atlas (Schafer 200-17 in our case) and iteratively refines each individual's parcel boundaries relative to their resting-state connectivity data[149]. It then computes concentration matrices (inverse covariance/partial correlation) for the whole sample via a group sparsity constraint. To ensure stability and precision, the algorithm iterates between the former and latter step for a total of 20 iterations or until no more than one vertex is changing per parcel[149]. For more details about GPIP, the reader is referred to our previous publication where we used GPIP on this sample[69]. Regional high kappa timeseries from ME-ICA were used to derive measures of local and global BOLD signal variability.

## fMRI young sample 2

A total of 112 healthy young individuals age matched to young sample 1, ages 18−34 ($M_{age}$ = 23 years, $SD_{age}$ = 3 years, 54% $F$), from the Enhanced Nathan Kline Institute Rockland Sample[75] were included in this study. All analyses were carried out on the whole sample (combined sex data). All individuals in the study were right-handed, healthy, with no current or past neurological, psychiatric diagnosis[153]. All participants provided written informed consent and were compensated for their time. Research protocols were approved by the NKI Institutional Review Board.

We included one run of resting-state fMRI data acquired on a 3T Siemens Trio scanner using a multiband (factor of 4) EPI sequence

(TR = 1400 ms; TE = 30 ms; 65° flip angle; FOV = 224 mm; 64 axial slices; 2 mm isotropic voxels; 404 volumes). Participants were instructed to keep their eyes open during the scan[75]. Functional images were preprocessed in the same fashion as our previous publication[153]. Briefly, fMRI data were preprocessed using the data preprocessing assistant for resting-state fMRI advanced edition toolbox (DPARSF-A v4.3_170105; http://rfmri.org/DPARSF)[153] as follows: removal of first 5 volumes, despiking, realignment, normalization to MNI template, smoothing (6 mm FWHM), ICA denoising via ICA-FIX trained on $n$ = 24 participants, and nuisance regression (linear detrend, Friston 24 motion parameters, bandpass filter of 0.01−0.1 Hz). Data were lastly parcelled using the standard Schaefer 200-17 network solution[56]. Regional timeseries were used to calculate local and global BOLD signal variability.

## fMRI adult lifespan sample 1

A total of 154 healthy adults ages 20−86 ($M_{age}$ = 49 years, $SD_{age}$ = 19 years, 62% $F$) from the greater Toronto area were included in this study. All analyses were carried out on the whole sample (both female and male data together). All individuals in the study were screened to be healthy with no major neurological and/or psychiatric condition, they were right-handed, fluent English speakers, cognitively normal (mini-mental state examination > 26), and with normal or corrected-to-normal vision[19,20]. All participants provided written informed consent and were compensated for their time. Research protocols were approved by the Research Ethics Board at Baycrest Health Sciences Center.

Resting-state fMRI data were collected on a 3T Siemens Trio scanner using an EPI sequence (TR = 2000 ms; TE = 27 ms; 70° flip angle; FOV = 192 mm; 40 axial slices; 3 mm isotropic voxels; 297 volumes). Participants were instructed to keep their eyes open during the scan. Functional images were preprocessed in the same fashion as our previous publication, using the optimizing of preprocessing pipelines for neuroimaging software package (OPPNI) (available at https://github.com/strotherlab/oppni)[19]. Briefly, brain masks were created for each participant in FSL, functional volumes were co-registered, corrected for motion, detrended and temporally filtered within 0.01−0.1 Hz. ICA-FIX was then used to denoise the data, trained on $n$ = 40 participants, and further nuisance regression was applied on the denoised timeseries (motion, white matter, vessels, CSF). Lastly, data were smoothed (7 mm FWHM), normalized to MNI space, and resampled to 4 $mm^3$ isotropic voxels. Data were parcelled using the standard Schaefer 200-17 network solution[56]. Regional timeseries were used to calculate local and global BOLD signal variability.

## fMRI adult lifespan sample 2

A total of 154 healthy adults, age and sex-matched to our lifespan sample 1, ages ($M_{age}$ = 49 years, $SD_{age}$ = 19 years, 62% $F$), from the Enhanced Nathan Kline Institute Rockland Sample[75] were included in this study. All analyses were carried out on the whole sample (combined sex data). All participants provided written informed consent and were compensated for their time. Research protocols were approved by the NKI institutional review board. Refer to fMRI young sample 2 for details about the resting-state data.

## fMRI-MEG sample

Within-individual fMRI-MEG data were obtained from the openly available stage II of the Cambridge Centre for Aging and Neuroscience (Cam-CAN) data repository[79,80]. We included resting-state MEG data from a total of 104 healthy young adults in the same age range 18−34, as our primary fMRI datasets ($M_{age}$ = 28 years, $SD_{age}$ = 4 years, 56% $F$). All individuals in the study were screened to be healthy with no major neurological and/or psychiatric condition, they were right-handed, and fluent English speakers[79,80]. All MEG analyses, except for fMRI-MEG local signal variability correlations, were run on this sample. All

analyses were carried out on the whole sample (combined sex data). fMRI-MEG associations were conducted on 103 out of the 104 individuals, that is on those subjects with available resting-state MEG and movie-watching fMRI data. All individuals provided written informed consent and were compensated for their time. Research was conducted in compliance with the Helsinki Declaration and was approved by the Cambridgeshire 2 Research Ethics Committee.

**Resting-state MEG data acquisition and processing.** All participants underwent an approximately 8-min resting-state eye-closed MEG and a structural T1 MRI. MEG data were collected from a 306-channel VectorView MEG system (Elekta Neuromag, Helsinki) with 102 magnetometers and 204 orthogonal planar gradiometers at 1000 Hz sampling rate. The head position of participants was continuously monitored using four head-position indicator coils, while ocular (EOG) and cardiac (ECG) external electrodes were used to monitor physiological artifacts. See[79,80] for details on the dataset and data acquisition.

MEG data were preprocessed using Brainstorm[81]. The preprocessing pipeline followed previous published work[154]. Line noise artifact (50 Hz; with 10 harmonics) were removed using a bank of notch filters, in addition to 88 Hz noise—a common artifact characteristic to the Cam-CAN dataset. Slow-wave and DC-offset artifacts were removed with a high-pass FIR filter with a 0.3-Hz cut-off. Signal-space projectors were used to remove cardiac artifacts and attenuate low-frequency (1–7 Hz) and high-frequency noisy components (40–400 Hz) due to saccades and muscle activity. The MRI volumes of each participant were automatically segmented using Freesurfer[155] and co-registered to the MEG recording using approximately 100 digitized head points.

We constrained brain source models for each participant to their individual T1-weighted MRI data. We computed head models for each participant using the Brainstorm overlapping-spheres approach, and cortical source models using the Brainstorm implementation of linearly constrained minimum-variance beamforming (2018 version for source estimation processes); both processes were run using default parameters. MEG source orientations at 15,000 vertices were constrained normal to the cortical surface. We projected the resulting MEG source model for each participant onto the default anatomy of Brainstorm (ICBM152). We down-sampled the cortical source maps by averaging the time series within each parcel of the standard Schaefer 200-17 atlas[56]. This resulted in a matrix of 200 regions by 150 s of scouted MEG time series data per person. Regional timeseries data were used to calculate local MEG variability.

We computed power spectral density (PSD) estimates across all vertices of the source model using the Welch method with 2 s windows of 50% overlap. This resulted in PSD with a frequency resolution of 1/2 Hz. We down-sampled the PSD estimates to the Schaefer atlas by taking the mean spectral density within each ROI. We parametrized the resulting neural power spectra from 1 to 40 Hz using the specparam algorithm as implemented in Brainstorm[81] with the following parameters: peak width limits [0.5 12]; maximum number of peaks: 3; minimum peak amplitude: 0.3 a.u.; peak threshold: 2.0 SDs; proximity threshold: 2.0 SDs; aperiodic mode: fixed. Finally, we extracted the aperiodic component from the resulting spectral models at each parcel, for each participant (i.e., 1/f exponent)[89].

**Movie-watching fMRI data acquisition and processing.** Participants were shown a shortened 8-min version of the "Bang! You're Dead" episode from the TV show "Alfred Hitchcock Presents" (1961). None of them had seen the movie before. Participants were told to pay attention to the movie while lying still in the scanner. Data were acquired on a Siemens 3T TIM Trio scanner using a 32-channel head coil. Functional data were collected with a multi-echo T2* EPI sequence with the following details: 193 total volumes, 32 axial slices, 3.7 mm thick, 0.74 mm gap, TR = 2470 msec, TE = [9.4, 21.2, 33, 45, 57] msec, flip angle = 78°, FOV = 192 × 192 mm, voxel size = 3 × 3 × 4.44 mm. Structural T1-weighted images were also used in this study for co-registration purposes. These latter images were acquired with a 3D MPRAGE sequence with the following details: TR = 2250 msec, TE = 2.99 msec, TI = 900 msec, flip angle = 9°, FOV = 256 × 240 × 192 m, 1 mm isotropic voxels, GRAPPA acceleration factor = 2.

Functional data were preprocessed using fMRIPrep v21.0.1[156] (minimal preprocessing) and tedana v2.5[157] for denoising (optimal combination of echoes, PCA using the "stabilized" Kundu decision tree, ICA to decompose BOLD TE-dependent and non-BOLD TE-independent signals)[158].

## Neurobiological data
To contextualize local and global BOLD signal variability with multiscale descriptions of brain organization, we relied on multiple open-source repositories and toolboxes.

## Microscale data
We included the principal axes of cytoarchitectural and microstructural differentiation obtained via a nonlinear manifold learning technique called diffusion map embedding[159,160], on histological staining of a ex vivo human brain and in vivo quantitative T1 (qT1) data.

The first gradient of cytoarchitectural differentiation was obtained from ultrahigh-resolution histological information from the BigBrain dataset[50]. BigBrain is a three-dimensional model of an adult human brain (Caucasian male, age 65), reconstructed from 20-mm-thick slices of a coronally-sectioned, Merker-stained post-mortem specimen. The profile of staining intensity along the thickness of the cortical sheet is thought to capture the composition of local cellular assemblies. To assure coverage along the thickness of the cortical sheet, cellular staining intensity was sampled and averaged across equivolumetric intracortical surfaces at every voxel step for (100-mm voxels) across 163,842 vertices per hemisphere. The resulting staining intensity surface was then parcellated using an anatomical atlas-constrained clustering approach, which produced a 1012-nodes solution for the BigBrain specimen, constrained by the Desikan–Killiany and Destrieux atlas boundaries. Pairwise product-to-moment correlation of nodal staining intensity profiles resulted in the affinity matrix used for gradient embedding.

The first gradient of microstructural differentiation was obtained from qT1 intensities sampled from and averaged across 50 healthy young participants ($M_{age}$ = 29.54 years, $SD_{age}$ = 5.62 years, 46% $F$) in an openly available dataset (MICA-MICs)[51]. All individuals were right-handed and without a history of neurological and/or psychiatric conditions. All analyses were carried out on the whole sample (combined sex data). All participants provided written informed consent and were compensated for their time. Research protocols were approved by the Ethics Committee of the Montreal Neurological Institute and Hospital. The profile of qT1 intensity along the thickness of the cortical sheet is shown to capture the variation in myelination in cortical regions. To assure coverage along the thickness of the cortical sheet on qT1 volumes, 14 equivolumetric intracortical surfaces were generated for each individual and combined into vertex-wise averages. Intensity profile maps were then parcellated according to the Desikan–Killiany and Destrieux atlas and averaged across 50 healthy young adults. Cross-correlation of nodal qT1 intensity profiles computed with partial correlation, with whole-cortex intensity profile controlled for, followed by log-transformation, resulted in the affinity matrix used for gradient embedding[51].

These two principal gradients of cytoarchitectural and microstructural differentiation were released with the BigBrainWarp toolbox in BigBrain native volume[161]. We resampled both gradients from BigBrain native volume to the standard volume defined by the 1 mm isotropic ICBM 2009c Nonlinear Asymmetric brain template[162] using linear interpolation. We finally parcellated both gradients using the standard Schaefer 200-17 parcellation solution[56].

 

**Table 1 | Breakdown of neurotransmitter receptor effects and types included in this study**

| Neurotransmitter receptor | Effect | Type |
|---|---|---|
| a4b2 | Excitatory | Ionotropic |
| M1 | Excitatory | Metabotropic |
| D1 | Excitatory | Metabotropic |
| D2 | Inhibitory | Metabotropic |
| GABAa | Inhibitory | Ionotropic |
| GABAa-bz | Inhibitory | Ionotropic |
| mGluR5 | Excitatory | Metabotropic |
| NMDA | Excitatory | Ionotropic |
| 5HT1a | Inhibitory | Metabotropic |
| 5HT1b | Inhibitory | Metabotropic |
| 5HT2a | Excitatory | Metabotropic |
| 5HT4 | Excitatory | Metabotropic |
| 5HT6 | Excitatory | Metabotropic |

Equally from BigBrainWarp, layer thickness data from the Big-Brain specimen for all six cortical layers were additionally included in this study. Briefly, staining intensity profiles were obtained from curvature-adjusted equidistant sampling at 200 points along the thickness of the cortical sheet. Layer transitions were identified by a convolutional neural network guided by expert neuroanatomists[163]. We processed and parcellated thickness maps for each layer as above. Thickness values were averaged across layers I–III to obtain supragranular estimations and across layers V–VI for infragranular estimations. Layer IV thickness was referred to as granular in the text.

### Mesoscale data

We used openly available neurotransmitter receptor data from the Neuromaps toolbox[47]. Specifically, we included PET-derived receptor density distributions for the following neurotransmitter systems: serotonin (5-HT1a, 5-HT1b, 5-HT2a, 5-HT4, and 5-HT6)[164], dopamine (D1[165], D2[166]), GABA (GABAa, GABAbz)[167], glutamate (mGluR5)[168], and acetylcholine (a4b2[169], M1[170]). In addition to the receptor data from Neuromaps, we also include a map of NMDA receptor density, fetched from https://github.com/netneurolab/hansen_receptors[171–173]. All maps were parcellated with the standard Schaefer 200-17 parcellation solution[56] using Neuromaps.

Given the heterogeneity in the molecular and chemical composition of each brain region[76], parcellated neurotransmitter receptor maps were used to compute the following regional composite scores: receptor diversity, excitation/inhibition ratio, ionotropic density and metabotropic density. Receptor diversity was estimated via the normalized Shannon entropy $H$ derived as shown in Eq. 1. $d$ indicates a region's normalized receptor composition, a vector wherein each value reflects each receptor's density value normalized to its maximum value across brain regions. $L$ is the total number of receptors, 13. Greater $H$ values signify greater receptor diversity for a brain area. Regional excitation/inhibition ratio was computed as the ratio between the mean density of excitatory receptors and the mean density of inhibitory receptors for each region (see Table 1 for a breakdown of receptor effects and types). Regional ionotropic receptor density was calculated as the mean density of ionotropic receptors within each area. Similarly, regional metabotropic receptor density was obtained as the mean density of metabotropic receptors within each region.

$$H = -\frac{sum(d * log(d))}{log(L)} \qquad (1)$$

We computed the first principal component of gene expression from the Allen human brain Atlas[52], what we referred in the study as transcriptional/molecular gradient. Regional microarray expression data were obtained from 6 post-mortem brains (1 female, ages 24−57 years, $M_{age} = 42.5$ years, $SD_{age} = 13.38$ years) and were processed using the Abagen toolbox[53,174,175]. To ensure complete coverage across both hemispheres, we mirrored samples bilaterally and interpolated missing voxels using a nearest-neighbor approach. All other parameters in Abagen were set to default. Finally, we only retained genes with differential stability >= 0.1, as a means of filtering out genes with high variability across donors[176]. Altogether, 15,633 genes were used in the principal component analysis. Note that we followed the same processing pipeline as in ref. 177, except we used the Schaefer 200-17 parcellation solution[56].

### Macroscale data

We retrieved macroscale brain maps available through the Neuromaps toolbox[47] and parcellated them with the standard Schaefer 200-17 network atlas[56]. These maps include: PET-derived maps of oxygen metabolism, glucose metabolism, cerebral blood flow, and cerebral blood volume[178]; large-scale gradients of brain organization, specifically the sensory-association axis[55], the principal gradient of fMRI static functional connectivity[54], and the MEG-derived intrinsic timescale[179]. Lastly, we calculated temporal autocorrelation scores on our two primary fMRI datasets by taking the product-to-moment correlation between successive timepoints (lag-1) and alternate (lag-2) timepoints of each regional BOLD timeseries tailored to each sample's TR, that is, lag-1 autocorrelation for young sample 1 and lag-2 for young sample 2, and created group-level spatial maps.

### Local signal variability estimation

**Local fMRI BOLD signal variability.** Local BOLD signal variability was estimated for every individual within each main fMRI sample (young samples 1 and 2). Denoised regional BOLD timeseries were first mean-centered to a whole-brain mean of 0. Moment-to-moment temporal variability of the BOLD signal was then calculated by taking the rMSSD[57] of each normalized regional timeseries, as shown in Eq. 2, where $x$ represents a region's BOLD signal intensity at two successive timepoints $i$ and $i+1$, and $n$ is the total number of timepoints for that region. At each region, we took the square root of MSSD to preserve its original units (MSSD is equivalent to variance, rMSSD to SD). Greater rMSSD values are indicative of regions with greater local variability levels.

$$rMSSD = \sqrt{\frac{\sum_{i=1}^{n-1}(x_{i+1} - x_i)^2}{n-1}} \qquad (2)$$

**Local MEG signal variability.** The same formula as above was used to derive the regional moment-to-moment temporal variability of MEG timeseries via rMSSD.

### Global fMRI BOLD signal variability estimation and validation

**Estimation.** Global BOLD signal variability was estimated via dynamic functional connectivity for every individual within each fMRI sample (young samples 1 and 2, lifespan samples 1 and 2). First, each regional BOLD timeseries was partitioned in equally sized windows via a sliding window approach within each sample. Following recent guidelines[180] to ensure reliable estimations, window width was chosen to comprise between 20 and 40 timepoints (TRs), window length was between 45 and 60 s, window shape was set to squared, Leonardi high pass filtering was applied[181], and windows were shifted by 1 TR. Given the heterogeneity in sampling rate across fMRI samples, this procedure resulted in a different number of timepoints per window and in a different number of total windows across datasets (young sample 1: 60 s window, 20 TRs per window, 181 total windows; young sample 2: 45 s

**Table 2 | Breakdown of our windowing approach per sample and window size**

| | 45 s window | 60 s window | 80 s window | 100 s window |
|---|---|---|---|---|
| Young sample 1 run 1 | 15 TRs per window, 186 windows | 20 TRs per window, 181 windows | 27 TRs per window, 174 windows | 33 TRs per window, 168 windows |
| Young sample 1 run 2 | 15 TRs per window, 186 windows | 20 TRs per window, 181 windows | 27 TRs per window, 174 windows | 33 TRs per window, 168 windows |
| Lifespan sample 1 | 23 TRs per window, 275 windows | 30 TRs per window, 268 windows | 40 TRs per window, 258 windows | 50 TRs per window, 248 windows |
| Lifespan sample 2 | 32 TRs per window, 368 windows | 43 TRs per window, 357 windows | 57 TRs per window, 341 windows | 71 TRs per window, 329 windows |

window, 32 TRs per window, 368 total windows; lifespan sample 1: 45 s window, 23 TRs per window, 275 windows; lifespan sample 2: 45 s window, 32 TRs per window, 368 total windows). For each window and dataset, we calculated functional connectivity measures per region pair, as their product-to-moment correlation. $NxNxT$ functional connectivity data tables for each individual were thus derived, where $N$ is the number of regions and $T$ the number of windows. Unlike most dynamic connectivity approaches that apply clustering methods on the data tables, in this study, we wanted to minimize additional user input and maximize data fidelity. We therefore introduced covSTATIS to derive dynamic functional connectivity, an approach that does not rely on clustering analyses.

As a three-way extension of multidimensional scaling and principal component analysis, covSTATIS is a multi-table method that linearly combines multiple similarity matrices (i.e., correlation/covariance tables) and uses eigenvalue decomposition to identify predominant structured patterns both at the group and individual level[58–60]. In our case, we used a combination of sliding window approaches and covSTATIS to examine, for each individual, the similarity of their windowed functional connectivity matrices over time and derive individual-level estimates of regional functional connectivity dynamics. First, we assessed, via the $R_v$ similarity coefficient (i.e., squared product-to-moment correlation)[65], the similarity across all $Y$ data tables across all individuals within each sample and stored these values in a $YxY$ $R_v$ matrix. We then took the first eigenvector of the $R_v$ matrix to generate weights for each connectivity table and used these weights to build an $NxN$ group compromise space—the linear combination of all matrices where regional connections more similar across time and individuals were given a higher weight since they were most represented in the sample.

We then submitted the group compromise space to eigenvalue decomposition and obtained a multivariate connectivity space, wherein regions that showed similar connectivity values over time were closer together than regions with less similar connectivity values across windows. covSTATIS next allowed us to back-project into this abstract multivariate Cartesian space, for every individual, each region's mean connectivity value over time across all windows and around it, the region's connectivity value for each window. Our last step involved calculating, for each individual, the area of the hull around each region's mean connectivity over time. Each convex hull was peeled on 95% of the data to control for outliers, similarly to traditional dynamic functional connectivity approaches[64]. Global BOLD variability thus corresponds to the regional area of the hull values: a greater area of the hull indicates greater distance, hence spread, in connectivity across windows for a specific region, and is therefore characteristic of regions with greater global BOLD variability.

Importantly, covSTATIS is not unlike other dynamic functional connectivity approaches, where window size selection depends on TR and scan length. We therefore advise the reader to follow the guidelines already put forward by others in this space[180], as they apply to all methods that rely on sliding window approaches. To reiterate, the advantage of covSTATIS over other dynamic functional connectivity methods lies in the minimal number of user-dependent decisions post sliding window estimation (e.g., covSTATIS does not require any

clustering). We refer the reader to our covSTATIS-dedicated publication where we contextualize our method, comprehensively walk through all its details, and offer a tutorial to facilitate its implementation using Open Data[60].

**Validation**. To validate covSTATIS as a robust dynamic functional connectivity method, we partitioned regional timeseries data from our young sample 1 (for both run 1 and 2) and our two lifespan samples into windows of 45, 60, 80 and 100 s, and used covSTATIS to derive global BOLD signal variability scores for each iteration and sample. Table 2 provides a detailed breakdown of our windowing procedure per sample and window size. After obtaining global BOLD signal variability scores, we used these measures to assess (1) covSTATIS' consistency across window lengths (all samples); (2) its test-retest reliability across runs for each window length (young sample 1, runs 1–2); and (3) its ability to robustly capture age-related alterations in dynamic functional connectivity across window lengths (lifespan samples 1–2). All our validation analyses were carried out both at the individual and at the group level.

For each sample, covSTATIS' consistency was evaluated at the individual level via intraclass correlation coefficients (ICC) and their 95% confidence intervals based on a mean-rating ($k = 4$ window lengths), consistency, two-way mixed effects model with window size as fixed effect and individuals as random term[182]. At the group level, consistency was assessed by relating in a pairwise fashion window-specific covSTATIS results across brain regions via Spearman rho statistics. At the individual level, covSTATIS' test-retest reliability was determined for each window length via Fisher-z transformed Spearman rho correlations derived within subject, then averaged across individuals and finally converted back to sample-average rho estimates. We additionally performed t-tests contrasting Fisher-z transformed correlations against zero and reported t-test values, along with their confidence intervals and Cohen's $d$ effect sizes. At the group level, covSTATIS' test-retest reliability was instead assessed by correlating group-level covSTATIS scores across regions between runs.

As a final validation step, we implemented a multivariate analysis technique, PLS[67,68,183], on our two lifespan samples, to test whether covSTATIS area of the hull measures were sensitive to known age-related alterations in dynamic functional connectivity across the adult lifespan. In other words, we interrogated the covariance between covSTATIS measures and age, and we validated our results across windows of different lengths. Specifically, we ran two PLS analyses: (1) contrasting covSTATIS' area of the hull scores calculated on 45 s windows for both lifespan samples (i.e., a 2-group 1-condition design, these are the main results reported in Fig. 1C); (2) contrasting covSTATIS values for all four window lengths across the two lifespan samples (i.e., a 2-group 4-condition design, these results are shown in Supplementary Fig. S2).

Briefly, PLS calculates a covariance matrix between two (or more) sets of variables. This covariance matrix undergoes singular value decomposition and, as a result, orthogonal latent variables are generated (LVs; similar to principal components in PCA), which explain the covariance between the sets of measures. Resulting LVs consist of a left

singular vector (*U*) of age weights, a right singular vector (*V*) of brain regions, and a diagonal matrix of singular values (*S*). Each element of *V*, also called loading/salience, represents the contribution of each region to each LV. To identify significant LVs, singular values were permuted 1000 times. For each individual, we obtained an estimate of the degree to which they expressed a particular LV's spatial pattern (brain score) by multiplying each regional loading by their original value and summing over all brain regions. We then correlated subject-level brain scores with age to establish the age contribution to the observed spatial pattern. Significant correlations were identified by bootstrapping with 1000 resamples, the correlation values, and generating 95% confidence intervals around the original correlation values. Lastly, to determine the significance of brain regions to each LV, we applied 1000 bootstrap resamples on the regional loadings. A *t*-like statistic was derived, the bootstrap ratio (BSR), which is the ratio of each regional weight to its bootstrapped standard error. We applied a threshold to BSRs at a value of ±2.

### Inter-sample reliability of local and global fMRI BOLD variability

To evaluate the reliability of local and global BOLD variability, we contrasted local and global BOLD signal variability between our two main fMRI samples, young samples 1 and 2. We calculated reliability across brain regions at the group level, and within each brain region and network across samples. Group reliability across regions was derived via product-to-moment correlation between samples for each metric. Regional- and network-level reliability were instead estimated via independent t-tests, one test per region and one per network. This allowed us to assess mean differences in local and global BOLD variability levels across samples for each region and network. Reliability was determined at $p > 0.05$.

### Local and global fMRI BOLD variability topographies

To examine the spatial organization of local and global BOLD variability for each of our main fMRI samples, we calculated group-level local and global BOLD variability for the whole brain and for each functional network separately. Despite using the Schaefer 200-17 parcellation solution[56], we decided to reduce the number of networks from 17 to 7 to facilitate interpretation, by merging together regions from different subnetworks into their principal network (e.g., visual central and peripheral into visual).

As a final step, we correlated group-level local and global BOLD variability spatial maps for each main fMRI sample, with 4 different spatial maps obtained on neurobiological data: the first gradient of cytoarchitectural differentiation, microstructural differentiation, transcriptional/molecular differentiation and static functional connectivity differentiation (details in previous sections). To test for significance, we applied 10,000 Hungarian spins on the regional labels of local and global BOLD variability spatial maps[184]. This allowed us to obtain 10,000 null models, while preserving spatial autocorrelation, to compare our original correlations against.

### Multiscale analyses

We assessed the multiscale neurobiological correlates of local and global BOLD variability, separately for our two primary fMRI datasets, by relating group-level local and global BOLD variability maps to micro-, meso-, and macro-scale variables described in previous sections. We computed the product-to-moment correlation, across brain regions, between local and global BOLD variability and each neurobiological variable. To evaluate the significance of the correlations, we applied 10,000 Hungarian spins on the regional labels of local and global BOLD variability. Results are visualized in text as annoted heatmaps. For an alternative visualization, we entered all correlation values, separately for each fMRI sample, into the Fruchterman Reingold layout spring embedding algorithm and set an absolute threshold of 0.3.

To quantify inter-sample convergence, we ran Pearson's product-to-moment correlations on the Fisher-z transformed correlations vectors characterizing the relationships between local and global BOLD variability, and multiscale variables for each fMRI sample.

To test for the central role of local and global BOLD variability in brain organization, we computed cartographical analyses, similarly to[39,85], on our sample-specific multiscale correlation matrices. We first assigned (1) local and global BOLD variability, (2) microscale variables, (3) mesoscale and (4) macroscale measures to four different communities. We then took the absolute value of the reported correlations and calculated, for each fMRI sample, the participation coefficient of local and global BOLD variability. Briefly, participation coefficient scores allowed us to determine how evenly distributed across spatial scales the correlations of local and global BOLD variability were, for each fMRI data type: the closer the score to 1, the greater their multiscale participation[86].

Finally, to assess the unique contribution of each multiscale variable in predicting local and global BOLD variability, we ran a DA per each metric and sample. DA was run, predicting local and global BOLD variability from all multiscale variables at once. DA allowed us to estimate the relative importance of each predictor in a single multiple regression model. DA contrasts two predictors at a time against all possible sub-models ($2^p$-1 sub-models, *p* being the total number of predictors) and quantifies the incremental contribution of each predictor when added to each subset of the remaining predictors, as the increase in $R^2$. In our study, we derived percentage scores for each predictor, indicating their unique contribution to the prediction model[87].

### Multimodal analyses

Similarly to how we characterized the topography of local and global fMRI BOLD variability, we delineated the spatial organization of group-level local MEG variability for the whole brain and for each functional network separately. We also obtained individual-level local MEG variability scores and 1/f exponent measures per network, akin to our fMRI analyses (see previous section). For each subject, we then averaged both estimates across brain regions to obtain whole-brain local MEG variability and 1/f exponent per person, that we could then correlate to each other. To expand on this whole-brain analysis, we built a hierarchical linear model where we accounted for regional heterogeneities in local MEG variability and 1/f exponent values (Fig. S7). Each region was entered in the model as a random effect to investigate region-level relationships between local MEG variability and the 1/f exponent.

Next, we simulated naturalistic neurophysiological time series at various arrhythmic motifs using the NeuroDSP toolbox[91]. Each simulation was 150 s long, linearly combined rhythmic and arrhythmic (i.e., periodic and aperiodic) components at random initial phases, and was sampled at 500 Hz. The simulated periodic component of each time-series consisted of a fixed alpha (peak frequency of 10 Hz, amplitude of 0.7 a.u., and band width of 2 Hz) and beta peak (peak frequency of 19 Hz, amplitude of 0.4 a.u., and band width of 5 Hz). We simulated 10 timeseries at various aperiodic slope parameters ranging from −0.7 to −1.5 in steps of −0.1. The parameters for simulations were selected based on previous literature[89]. We used these simulations to test for the linear relationship between the steepness of the 1/f exponent and local MEG variability.

Lastly, to investigate multimodal correlates of local brain variability, we related group-level spatial maps of local fMRI BOLD variability (young sample 1 and 2) to group-level spatial maps of local MEG variability, via product-to-moment correlations.

### Reporting summary

Further information on research design is available in the Nature Portfolio Reporting Summary linked to this article.

## Data availability

Resting-state fMRI data from young sample 1 can be accessed on OpenNeuro (ds003592, version 1.0.13) at the following root link: https://openneuro.org/datasets/ds003592. Resting-state fMRI data from young sample 2 and lifespan sample 2 are available for download at the following link: https://rocklandsample.org/accessing-the-neuroimaging-data-releases. Resting-state fMRI-MEG data can be accessed by requesting the data at: https://www.cam-can.org/index.php?content=dataset. Microscale neurobiological data are downloadable from the BigBrainWarp toolbox: https://bigbrainwarp.readthedocs.io/en/latest/pages/installation.html. Mesoscale and macroscale neurobiological data are retrievable from the Abagen and Neuromaps toolboxes: https://github.com/netneurolab/abagen; https://github.com/netneurolab/neuromaps. Source data are provided with this paper.

## Code availability

This manuscript relied on preprocessed fMRI and MEG data and used various open access resources. fMRI data from young sample 1 were preprocessed using ME-ICA v3.2 (https://github.com/ME-ICA/me-ica). fMRI data from young sample 2 and lifespan sample 2 were preprocessed using the DPARSF-A toolbox v4.3_170105 (http://rfmri.org/DPARSF). fMRI data from lifespan sample 1 were preprocessed using the Optimizing of Preprocessing Pipelines for Neuroimaging Software Package (OPPNI) (available at https://github.com/strotherlab/oppni). Movie watching fMRI data from the fMRI-MEG dataset were preprocessed using fMRIprep 21.0.1 and tedana v2.5. MEG data were preprocessed using Brainstorm (March 2021 distribution). All data analyses and the data presented in the figures, except for the electrophysiological timeseries simulations, were obtained in R v4.2-4.3 and MATLAB R2022b, R2024b. Partial Least Squares analyses were carried out using an openly available MATLAB toolbox (https://github.com/McIntosh-Lab/PLS/). Simulated naturalistic electrophysiological timeseries were generated using the NeuroDSP toolbox (https://neurodsp-tools.github.io/neurodsp). Custom code to reproduce data analyses can be at the following link: https://github.com/giuliabaracc/BiologicalVariability/tree/main.

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

## Acknowledgements

This research was supported in part by grants from the Canadian Institutes of Health. Research (CIHR), NIH (1S10RR025145; R01AG068563), and the Natural Sciences and. Engineering Research Council of Canada (NSERC). G.B. is supported by a Fonds de recherche du Québec-Santé (FRQS) Doctoral Fellowship. R.N.S. is a Research Scholar also supported by FRQS. L.Q.U. is supported by the National Institute of Child Health and Human Development (R21HD111805 and R01HD11669) and the National Institute on Drug Abuse (U24DA041147 and U01DA050987). Thanks to Drs. Colleen Hughes, Alfie Wearn and Prof. Mac Shine for the insightful discussions.

## Author contributions

G.B.: conceptualization, methodology, software, formal analysis, investigation, data curation, writing–original draft, writing–review and editing, visualization, funding acquisition. Y.Z.: formal analysis, visualization, writing–review and editing. J.d.S.C.: formal analysis, data curation, writing– review and editing. J.Y.H.: data curation, writing–review and editing. C.F.: data curation, writing– review and editing. R.S.: data curation, writing–review and editing. J.R.R: investigation, data curation, writing–review and editing. G.R.T.: resources, writing–review and editing. C.L.G.: resources, writing–review and editing. B.M.: writing–review and editing. J.S.N.: data curation, resources, writing–review and editing. L.Q.U.: conceptualization, resources, supervision, writing– review and editing, funding acquisition. R.N.S.: conceptualization, resources, supervision, writing– original draft, writing–review and editing, funding acquisition.

## Competing interests

The authors declare no competing interests.
