## [Transparent Peer Review File · Nature Communications]

The biological role of local and global fMRI BOLD signal variability in multiscale human brain organization

Corresponding Author: Dr Giulia Baracchini

Version 0:

Reviewer comments:

Reviewer #1

(Remarks to the Author)

This study showed that BOLD signal variability represents a spatially heterogeneous, central property of multi-scale multi-modal brain organization, which is distinct from noise and "rooted" in electrophysiological processes. Local BOLD signal variability is defined as the regional moment-to-moment change in BOLD signal intensity between successive timepoints while the global BOLD signal variability is defined as the similarity in inter-regional functional connectivity over time. The introduction is well-written, which establishes the rationale for this study. However, several methodological and interpretational issues mitigate my enthusiasm for the current findings.

1. The claim that "BOLD signal variability is rooted in electrophysiological processes" needs more evidence and support. First of all, we know that BOLD and MEG have different neural underpinnings. BOLD is related to local post-synaptic potentials. There is also neurovascular coupling to be taken into account. The current data in Figure 5C only demonstrate the negative association between group-level local BOLD signal variability and group-level local MEG signal variability via one spatial correlation (if I understood correctly). There is no discussion on this either. It would be more convincing to use concurrent fMRI-EEG data on the same group of subjects and show similar associations at the individual level.
2. It is interesting to see that covSTATIS-derived global BOLD signal variability has a high test-retest reliability of 0.98 using two successive runs. The sliding window size was chosen to be 45-60s. I wonder how this global variability measure, its reliability, and age effect change with window size from 40s to 100s, which is usually used in dynamic functional connectivity literature. It would be good to provide some guidance on the optimal window size for such a measure.
3. I also wonder why "global signal variability" rather than "connectivity variability" or something similar is used here. The measure is not exactly based on the BOLD signal itself but rather on the correlations between two time series, i.e., connectivity.
4. Discussion: "This statistical explanation elucidates why, in this study, lower-order regions exhibited lower reliability of variability measures, than higher-order areas across fMRI samples." It is quite confusing to come to this conclusion as all previous text on statistical explanation is related to local and global BOLD signal variability (e.g., which one corresponds closer to the fMRI data etc). No lower-order and higher-order brain regions were mentioned before this. Please clarify and discuss.
5. What are the unique features of specific low-level and high-level brain regions in the context of local and global BOLD signal variability? The word "heterogenous" is used frequently but there is a lack of discussion on the patterns common or unique to local and global signal variability.
6. It is nice to see a trend of how local and global BOLD variability relate to different gradients in young samples (Figure 3C). However, the level of evidence is a bit weak as some did not pass spin test in both young samples. Is there any reason behind this? For functional gradient, besides the principal unimodal-transmodal gradient, it might be informative to check the unimodal gradient (2nd one). Moreover, what is the pattern in the old sample? Do we see the same dissociation?
7. What is the additional value of the local and global BOLD signal variability proposed here on top of or in comparison with some other BOLD-based measures like static FC (the authors have found high correlation between static FC and global variability), ALFF, other DFC measures. It would be useful to demonstrate its behavioral relevance.

Reviewer #2

(Remarks to the Author)

- **Key results:** The manuscript examines the role of signal variability in brain function. Using several large fMRI datasets, the authors look at the association between variability and dynamic functional connectivity with microstructure, transcriptomics, neurotransmitter receptor and metabolic data, static connectivity, and empirical and simulated MEG data. They suggest that variability is structured and spatially heterogeneous, and is important in multi-scale characterizations of brain function.

- **Validity:** I do not see any fatal flaws, but the reasoning behind methodological choices and variables included needs to be better described. These choices are especially important as they can significantly impact outcomes in variability work (the authors themselves suggest this). Specifically, in the literature, variability and dynamic functional connectivity (dFC) tend to be treated somewhat differently. I see no issues with choosing a dFC-type measure to index “global variability”, but the reasoning behind the choice needs to be framed in the context previous literature in the paper. In addition, local and global complexity has been discussed quite a bit with EEG/MEG (see for example <https://pubmed.ncbi.nlm.nih.gov/21525281/>; <https://pubmed.ncbi.nlm.nih.gov/23395850/>). I am not sure why these papers/ideas are not discussed as they are relevant, and why, in the context of MEG, they don't appear to be considered in analytic choices. Is this because the focus is more on variability than complexity in signal?

- **Originality and significance:** Several papers have examined the importance/unique contribution of variability measures in understanding brain function (seminal papers include <https://www.pnas.org/doi/full/10.1073/pnas.0901831106>; <https://www.jneurosci.org/content/30/14/4914>; <https://www.ncbi.nlm.nih.gov/pmc/articles/PMC3922711/>). The value added in this manuscript is the impressive amount of data examined, the gamut of micro and macroscale variables that brain signal variability was linked to, and the examination of local and global variability in BOLD fMRI. To better highlight significance, the authors should focus on these unique contributions rather than the idea that variability is a useful metric. They should better motivate their choice of macro and microscale variables, their analysis methods (some which do not follow directly from previous literature), and include greater discussion of the implications of their specific findings (i.e., why and how is brain signal variability an important measure for brain characterisation? How can it fit with / add to other more commonly used measures? How should knowing that variability has “spatially heterogeneous topography, encapsulate micro-, meso- and macroscale neurobiological phenomena” influence the use of this measure in cognitive network neuroscience?).

- **Data & methodology:** The data and methodology are sufficiently described in text (with references to more detailed previously published descriptions) to enable reproducing the results. As a minor point, added details regarding Group Prior Individual Parcellation (GPIP), specifically how it may influence results over more commonly used parcellation methods would be useful.

- **Appropriate use of statistics and treatment of uncertainties:** The statistical tests appear to be performed appropriately.

- **Conclusions:** Data interpretation and related conclusions are valid.

- **Suggested improvements:** No additional data or experiments are necessary.

- **References:** Some of the references mentioned above need to be added. In addition:

- o The following sentence is missing references “The human brain is a complex stochastic system, hence integrating functional variability of brain signals is essential for the modeling of human brain function.” Deco et al's (2009) paper from above speaks to this and could be added.

- o The authors suggest that “computational approaches to brain dynamics may thus benefit from introducing spatial heterogeneity in local measures of variability, to more precisely approximate empirical observations.” Has this not been done? And if not, this is really interesting and would benefit from a little more discussion.

- o The finding that global variability increases with age could be better contextualised within the literature, as I believe this is the area in which fMRI signal variability analyses are most commonly performed. There variability papers already cited in the manuscript show consensus with overall increases in variability with age, but an acknowledgement that there are some papers that found general decreases or a mix of increases and decreases (e.g., <https://pubmed.ncbi.nlm.nih.gov/32915200/>; <https://pubmed.ncbi.nlm.nih.gov/35901557/>) could be included.

- **Clarity and context:** The abstract is clear and accessible but the introduction and discussion are overly concise, especially for an experimental design as complex as the one used in this paper.

Version 1:

Reviewer comments:

Reviewer #1

(Remarks to the Author)

I appreciate the hard work from the authors. The revision has addressed some concerns, but there are still three main concerns left:

1. What are the unique features of specific low-level and high-level brain regions in the context of local and global BOLD signal variability? The authors did not perform any additional analyses to compare them quantitatively. I think this is important even though they are derived independently. The newly added discussion is also non-specific and not satisfying. E.g. “cortical areas devoted to local processing presented variability patterns that were more dependent on fMRI data type for both local and global variability measures, than higher-order regions.” What are the cortical areas devoted to local processing? How is it different from higher-order regions? Given the rich literature on functional gradients and organization

involving sensorimotor and association regions, more comparison and interpretation for local and global variability is needed. To me, this response contradicts the authors' claim in their earlier response that these two processes are linked, and in this study they purposely named them differently from the literature to study both (single time-series variability and paired time series (connectivity) variability). Without fully addressing this point, providing additional insights for the field is hard.

2. Patterns in the sample with older adults. This question has not been addressed. I do not think it is unreasonable to repeat the same analyses in old samples.

3. Regarding my last comment on the relationship and comparisons with other measures, I am aligned with Reviewer 2 R2 that it is important to demonstrate why this metric is better than others and their implications in understanding our brain development, aging and behavior associations.

(Remarks on code availability)

Reviewer #2

(Remarks to the Author)

The authors have addressed all my concerns.

(Remarks on code availability)

Reviewer #1:

REVIEWER 1, SUMMARY: This study showed that BOLD signal variability represents a spatially heterogeneous, central property of multi-scale multi-modal brain organization, which is distinct from noise and "rooted" in electrophysiological processes. Local BOLD signal variability is defined as the regional moment-to-moment change in BOLD signal intensity between successive timepoints while the global BOLD signal variability is defined as the similarity in inter-regional functional connectivity over time. The introduction is well-written, which establishes the rationale for this study. However, several methodological and interpretational issues mitigate my enthusiasm for the current findings.

We thank the reviewer for their support for our study and their constructive feedback.

REVIEWER 1, COMMENT 1: The claim that "BOLD signal variability is rooted in electrophysiological processes" needs more evidence and support. First of all, we know that BOLD and MEG have different neural underpinnings. BOLD is related to local post-synaptic potentials. There is also neurovascular coupling to be taken into account. The current data in Figure 5C only demonstrate the negative association between group-level local BOLD signal variability and group-level local MEG signal variability via one spatial correlation (if I understood correctly). There is no discussion on this either. It would be more convincing to use concurrent fMRI-EEG data on the same group of subjects and show similar associations at the individual level.

RESPONSE TO R1, COMMENT 1: We appreciate the reviewer's positive feedback on our research question and approach, and we thank them for raising this point. As correctly noted, we claimed that fMRI BOLD variability is rooted in electrophysiological processes based on a significant spatial correlation between group-level fMRI and MEG variability measures. The reviewer is correct in that such measures were obtained in two different young adult samples. As the reviewer mentioned, fMRI and MEG signals arise from distinct neurobiological sources, making their association non-trivial. We agree that the use of concurrent fMRI-EEG data could therefore provide more robust, and clearer, evidence in support of this multimodal association, ultimately strengthening our claim. Given that the MEG dataset used in this work also had an fMRI component (Taylor et al., 2017 NeuroImage), we decided to use these openly available within-subject fMRI-MEG data in this revision to address the reviewer's comment. In sum, we used these data to empirically determine whether such an association is observable within-subject, rather than a single association between-group results.

Consistent with the between-group findings, we now show a similar and reliable negative relationship between fMRI BOLD variability and MEG variability at the within-subject level. While the magnitude of the group-level association is similar to the one reported in-text using two different samples ($r = -0.2$), the within-subject relationship is lower (average $r = -0.14$). Nonetheless, it still shows a large effect size (Cohen's $d = -1.17$). We have now included these new results, their interpretation and details on the dataset in our revised manuscript and here below.

Revised Results:

“As a final step, we tested cross-modal relationships between local fMRI and MEG signal variability. We report both cross-sample and within-sample associations (i.e., fMRI-MEG data obtained on those individuals from the MEG Sample with available fMRI data – $n=103$). At the group level, we found a consistent, negative association across and within sample between fMRI and MEG variability (**Figure 5C**). A similar negative association was also observed at the individual level (average Fisher z -to- $r = -$

0.14; $t(102) = -11.91$, $p < 0.001$, 95% C.I. = [-0.17, -0.12], Cohen's $d = -1.17$). Together, these findings reinforce the biological multimodal nature of local brain signal variability.” (p. 22)

A Functional topography of local MEG variability via rMSSD

B Functional topography of 1/f Exponent & relationship with local MEG variability

C Local fMRI BOLD variability relates to local MEG variability

Figure 1. Contextualizing local brain variability across neuroimaging modalities. | Local electrophysiological variability was estimated as MEG signal variability via rMSSD, and as the 1/f exponent in the MEG power spectrum. **(A)** Functional topography of local rMSSD-derived MEG variability. Central, grey-scale map represents the group average map. Around it are the 7 canonical network-level group maps. Note that rMSSD values were z-scored, as units are arbitrary. | **(B) Left:** Functional topography of the 1/f exponent per network, across individuals. **Middle:** whole-brain relationship between local rMSSD-derived MEG variability and 1/f exponent across individuals. **Right:** We simulated naturalistic electrophysiological timeseries and varied the steepness of their 1/f exponent at various parameters from -1.5 to -0.7, shown as positive values in the graph, in steps of -0.1. 10 simulations were run per parameter. We calculated local rMSSD-derived MEG variability on each simulated timeseries, and related exponent values with rMSSD scores. | **(C)** Cross-modal **spatial** relationships between local fMRI and MEG brain variability, **between two independent samples (left and middle figures) and within the same sample using fMRI-MEG data (right figure).**

Revised Discussion

[...] **Via empirical and simulated MEG data, we first** found local signal variability to be driven by the slope of the signal power spectrum [...].

Via cross-sample and within-sample comparisons, we observed a consistent negative association between fMRI-derived and MEG-derived local brain signal variability. These results expand on previous literature showing a spatial correspondence between fMRI and MEG signals using static functional connectivity¹³⁷⁻¹⁴¹. Our findings further demonstrate that fMRI and MEG signals share meaningful information about human brain dynamics. Here, the negative direction of fMRI-MEG correlations may be explained by differences in the topographies of the two signals, as they originate from distinct physiological sources and have different frequency content¹⁴²⁻¹⁴⁴. The relatively modest magnitude of their correlation may be due to differences in study design between the two modalities: MEG data were acquired during rest, while fMRI data during movie watching⁷⁹. While further work is needed to shed more light on the nature of these relationships, our work provides preliminary evidence in favor of the biological multimodal nature of local brain signal variability. Specifically, these findings urge the general neuroscience community to re-consider what is empirically deemed “noise” (i.e., slope of the electrophysiological power spectrum, local BOLD signal variability) as biologically meaningful properties of multimodal brain signals. (pp. 28-29)

Revised Methods:

fMRI-MEG Sample

Within-individual fMRI-MEG data were obtained from the openly available Stage II of the Cambridge Centre for Aging and Neuroscience (Cam-CAN) data repository^{79,80}. We included resting-state MEG data from a total of 104 healthy young adults in the same age range 18-34 as our primary fMRI datasets ($M_{\text{age}} = 28\text{y}$, $SD_{\text{age}} = 4\text{y}$, 56% F). All MEG analyses, except for fMRI-MEG local signal variability correlations, were run on this sample. fMRI-MEG associations were conducted on 103 out of the 104 individuals, that is on those subjects with available resting-state MEG and movie-watching fMRI data. All individuals provided written informed consent. Research was conducted in compliance with the Helsinki Declaration and was approved by the Cambridgeshire 2 Research Ethics Committee.

Movie-watching fMRI data acquisition and processing

Participants were shown a shortened 8-minute version of the “Bang! You’re Dead” episode from the TV show “Alfred Hitchcock Presents” (1961). None of them had seen the movie before. Participants were told to pay attention to the movie while lying still in the scanner. Data were acquired on a Siemens 3T TIM Trio scanner using a 32-channel head coil. Functional data were collected with a multi-echo T2* EPI sequence with the following details: 193 total volumes, 32 axial slices, 3.7 mm thick, 0.74 mm gap,

TR = 2470 msec, TE = [9.4, 21.2, 33, 45, 57] msec, flip angle = 78°, FOV = 192 × 192 mm, voxel size = 3 × 3 × 4.44 mm. Structural T1-weighted images were also used in this study for co-registration purposes. These latter images were acquired with a 3D MPRAGE sequence with the following details: TR = 2250 msec, TE = 2.99 msec, TI = 900 msec, flip angle = 9°, FOV = 256 × 240 × 192 mm, 1 mm isotropic voxels, GRAPPA acceleration factor = 2.

Functional data were preprocessed using fMRIPrep 21.0.1¹⁵⁶ and tedana¹⁵⁷, as reported in our previous publication¹⁵⁸. (pp. 33-35)

REVIEWER 1, COMMENT 2: It is interesting to see that covSTATIS-derived global BOLD signal variability has a high test-retest reliability of 0.98 using two successive runs. The sliding window size was chosen to be 45-60s. I wonder how this global variability measure, its reliability, and age effect change with window size from 40s to 100s, which is usually used in dynamic functional connectivity literature. It would be good to provide some guidance on the optimal window size for such a measure.

RESPONSE TO R1, COMMENT 2: We thank the reviewer for their comment, which has prompted us to write a separate paper—now published in Aperture Neuro (Baracchini et al., 2024 Aperture Neuro)—about covSTATIS as a new tool for cognitive network neuroscience.

To address the reviewer’s specific comment on window size, we assessed the consistency and reliability of covSTATIS-derived global BOLD signal variability results using sliding windows of different lengths. Specifically, we partitioned regional timeseries data from our Young Sample 1 (for both run 1 and 2) and our two Lifespan Samples into windows of 45sec, 60sec, 80sec and 100sec, and used covSTATIS to derive global BOLD signal variability scores for each iteration and sample. We used these global BOLD signal variability scores to assess (1) their consistency across window lengths (all samples); (2) their test-retest reliability across runs for each window length (Young Sample 1, runs 1-2); and (3) the stability of their age effects across window lengths (Lifespan Samples 1-2).

We found covSTATIS-derived global BOLD variability scores to be (1) highly consistent across window lengths both at the individual level (intraclass coefficient estimates and their 95% C.I.s obtained via a mean-rating (k=4 window lengths), consistency, two-way mixed effects model with window size as fixed effect and individuals as random term: ICC > 0.82 for all samples; **Supplemental Figure S1**) and group level (Spearman rho between pairwise group-level window results across regions: $0.92 < r < 0.99$, $p < .001$) in all four samples; (2) moderately reliable at the individual level for all window lengths (average $r \sim 0.30$, $p < .001$, large Cohen’s $d > 1.6$); and (3) highly stable in their age effects across window lengths for both aging samples (PLS analysis with 4 conditions and 2 groups: LV1, at $p < .001$ with 58% variance explained, largely recapitulating Figure 1C results both in terms of location of effects and their directionality; **Supplemental Figure S2**). We report below, and add to our main text, all details about the results and analyses. To note, these new reliability results (2) are lower than the ones reported in Figure 1C because they were obtained within individuals across regions, and not on averaged group values across regions (i.e., what we show in Figure 1C). Since all our analyses are conducted at the group level, we kept our group-level findings in text and added our individual-level reliability results to Supplemental.

Together, these analyses show that covSTATIS is a robust tool for global BOLD signal variability (i.e., dynamic functional connectivity) estimations. While we agree with the reviewer that it would be beneficial to provide guidance on the optimal window size for covSTATIS, covSTATIS is not unlike other dynamic functional connectivity approaches, where optimal window size selection depends on TR and scan length. We therefore advise the reader to follow the guidelines already put forward by others in this space (Liégeois et al., 2017 NeuroImage), as they apply to all methods that rely on sliding window

approaches. To reiterate, the advantage of covSTATIS over other dynamic functional connectivity methods lies in the minimal number of user-dependent decisions *post sliding window estimation* (e.g., covSTATIS does not require any clustering). In this revision, we refer the reader to our new covSTATIS publication where we contextualize our method, comprehensively walk through all its details, and offer a tutorial to facilitate its implementation using Open Data. Together, we appreciate the reviewer's curiosity about the method, which prompted us to provide the details of our approach in an appropriate venue for a fulsome presentation (which was outside of the scope of the present manuscript).

Revised Results

“Global BOLD signal variability was estimated as dynamic functional connectivity using covSTATIS. As a three-way extension of Multidimensional Scaling and Principal Component Analysis, covSTATIS is a multi-table method that linearly combines multiple similarity matrices (i.e., correlation/covariance tables) and uses eigenvalue decomposition to identify predominant structured patterns both at the group and individual level⁵⁸⁻⁶⁰. In our case, we used a combination of sliding window approaches and covSTATIS to examine, for each individual, the similarity of their windowed functional connectivity matrices over time and derive individual-level estimates of regional functional connectivity dynamics. Unlike most conventional fMRI dynamic connectivity methods⁶¹ that operate on coarser brain dynamics estimates, covSTATIS defines finer *regional* measures of global dynamics and does so with minimal user-dependent input, as it side-steps commonly adopted clustering approaches that have traditionally led to fractionated, study-specific definitions of dynamic functional connectivity⁶². For more information about covSTATIS and its implementation, we refer the reader to our dedicated recent publication⁶⁰.” (p. 5-6)

“We next validated our method by assessing its consistency, test-retest reliability and its ability to capture age effects typically reported in studies using traditional dynamic connectivity methods. We used resting-state fMRI data from one sample of 145 healthy young adults with two successive runs, and two independent cross-sectional healthy lifespan datasets (see Methods for details). For each sample and individual, we partitioned their regional timeseries data into windows of different length and derived covSTATIS’ area of the hull scores (i.e., global BOLD signal variability) for each iteration. We found global BOLD variability to be highly consistent across window lengths both at the individual level (two-way mixed effects ICC model > 0.82 for all samples; **Supplemental Figure S1**) and at the group level ($0.92 < \text{Spearman } \rho < 0.99$, $p < .001$) in all samples. Global BOLD variability additionally showed high reliability at the group level ($r=0.98$; $p<.001$; **Figure 1C**) and moderate reliability at the individual level for all window lengths (average $r \sim 0.30$, $p<.001$, large Cohen’s $d > 1.6$). In line with current literature demonstrating an age-induced dampening of the brain’s dynamic range⁶⁶, here we used Partial Least Squares^{67,68}—a multivariate method that assesses the covariance between two or more sets of variables—and showed that covSTATIS-derived area of the hull scores also decreased with age particularly in regions that preferentially show age effects in the literature⁶⁹ (**Figure 1C**; one significant latent variable (LV1) at $p=0.003$ explaining 72% brain-age variance; Lifespan Sample 1 brain-age $r=-0.19$, Lifespan Sample 2 brain-age $r=-0.30$). Furthermore, age effects were stable across windows of different length both in terms of their spatial location and directionality (LV1 at $p<.001$ explaining 58% brain-age variance; **Supplemental Figure S2**). Together, these results highlight how covSTATIS is a valid and robust tool to estimate global BOLD signal variability.” (p. 6-7)

Consistency of covSTATIS results across multiple window lengths

Young Sample 1, run 1

95% C.I.

Young Sample 1, run 2

95% C.I.

Networks

- Visual
- Somato-Motor
- Dorsal Attention
- Ventral Attention
- Limbic
- Control
- Default

Lifespan Sample 1

95% C.I.

Lifespan Sample 2

95% C.I.

Figure S1. CovSTATIS shows consistent results across windows of different length. We calculated global BOLD signal variability (i.e., covSTATIS’ area of the hull values) for each fMRI sample across windows of different length (45sec, 60sec, 80sec, 100sec; see Methods for full details). We assessed within-sample inter-window consistency via intraclass coefficient statistics (ICC) and their 95% confidence intervals. ICCs were derived via a mean-rating (k=4 window lengths), consistency, two-way mixed effects model with window size as fixed effect and individuals as random term. ICC values were greater than 0.82 for all samples, indicating high consistency of covSTATIS estimates.

Validation of covSTATIS age effects across multiple window lengths

Figure S2. CovSTATIS-derived age effects are stable across windows of different length. To assess the stability of covSTATIS age effects, we ran our PLS analyses contrasting covSTATIS values for four window lengths across the two Lifespan Samples (i.e., a 2-group 4-condition design; see Methods for full details). We found age effects to be stable both in terms of their spatial location (brain plot on the left) and directionality (bar plot on the right; LV1 at $p < .001$ explaining 58% brain-age variance).

Revised Methods

Global fMRI BOLD signal variability estimation & validation

Estimation

“[...] As a three-way extension of Multidimensional Scaling and Principal Component Analysis, covSTATIS is a multi-table method that linearly combines multiple similarity matrices (i.e., correlation/covariance tables) and uses eigenvalue decomposition to identify predominant structured patterns both at the group and individual level⁵⁸⁻⁶⁰. In our case, we used a combination of sliding window approaches and covSTATIS to examine, for each individual, the similarity of their windowed functional connectivity matrices over time and derive individual-level estimates of regional functional connectivity dynamics. First, we assessed, via the R_v similarity coefficient (i.e., squared product-to-moment correlation)⁶⁵, the similarity across all Y data tables across all individuals within each sample and stored these values in a $Y \times Y$ R_v matrix. We then took the first eigenvector of the R_v matrix to generate weights for each connectivity table and used these weights to build an $N \times N$ group compromise space—the linear

combination of all matrices where regional connections more similar across time and individuals were given a higher weight since they were most represented in the sample. [...]

Importantly, covSTATIS is not unlike other dynamic functional connectivity approaches, where window size selection depends on TR and scan length. We therefore advise the reader to follow the guidelines already put forward by others in this space¹⁸⁰, as they apply to all methods that rely on sliding window approaches. To reiterate, the advantage of covSTATIS over other dynamic functional connectivity methods lies in the minimal number of user-dependent decisions *post sliding window estimation* (e.g., covSTATIS does not require any clustering). We refer the reader to our covSTATIS-dedicated publication where we contextualize our method, comprehensively walk through all its details, and offer a tutorial to facilitate its implementation using Open Data⁶⁰.” (p. 39-40)

Validation

“To validate covSTATIS as a robust dynamic functional connectivity method, we partitioned regional timeseries data from our Young Sample 1 (for both run 1 and 2) and our two Lifespan Samples into windows of 45sec, 60sec, 80sec and 100sec, and used covSTATIS to derive global BOLD signal variability scores for each iteration and sample. **Table 2** provides a detailed breakdown of our windowing procedure per sample and window size. After obtaining global BOLD signal variability scores, we used these measures to assess (1) covSTATIS’ consistency across window lengths (all samples); (2) its test-retest reliability across runs for each window length (Young Sample 1, runs 1-2); and (3) its ability to robustly capture age-related alterations in dynamic functional connectivity across window lengths (Lifespan Samples 1-2). All our validation analyses were carried out both at the individual and at the group level.

Table 2. Breakdown of our windowing approach per window length and sample.

	45sec window	60sec window	80sec window	100sec window
Young Sample 1 run 1	15 TRs per window, 186 windows	20 TRs per window, 181 windows	27 TRs per window, 174 windows	33 TRs per window, 168 windows
Young Sample 1 run 2	15 TRs per window, 186 windows	20 TRs per window, 181 windows	27 TRs per window, 174 windows	33 TRs per window, 168 windows
Lifespan Sample 1	23 TRs per window, 275 windows	30 TRs per window, 268 windows	40 TRs per window, 258 windows	50 TRs per window, 248 windows
Lifespan Sample 2	32 TRs per window, 368 windows	43 TRs per window, 357 windows	57 TRs per window, 341 windows	71 TRs per window, 329 windows

For each sample, covSTATIS’ consistency was evaluated at the individual level via intraclass correlation coefficients (ICC) and their 95% confidence intervals based on a mean-rating (k=4 window lengths), consistency, two-way mixed effects model with window size as fixed effect and individuals as random term¹⁸². At the group level, consistency was assessed by relating in a pairwise fashion window-specific covSTATIS results across brain regions via Spearman rho statistics. At the individual level, covSTATIS’ test-retest reliability was determined for each window length via Fisher-z transformed Spearman rho correlations derived within subject, then averaged across individuals and finally converted back to sample-average rho estimates. We additionally performed t-tests contrasting Fisher-z transformed

correlations against zero and reported t-test values, along with their confidence intervals and Cohen's d effect sizes. At the group level, covSTATIS' test-retest reliability was instead assessed by correlating group-level covSTATIS scores across regions between runs.

As a final validation step, we implemented a multivariate analysis technique, Partial Least Squares (PLS)^{67,68,183}, on our two Lifespan Samples, to test whether covSTATIS area of the hull measures were sensitive to known age-related alterations in dynamic functional connectivity across the adult lifespan. In other words, we interrogated the covariance between covSTATIS measures and age, and we validated our results across windows of different lengths. Specifically, we ran two PLS analyses: (1) contrasting covSTATIS' area of the hull scores calculated on 45sec windows for both Lifespan Samples (i.e., a 2-group 1-condition design, these are the main results reported in Figure 1C); (2) contrasting covSTATIS values for all four window lengths across the two Lifespan Samples (i.e., a 2-group 4-condition design, these results are shown in **Supplemental Figure S2**).

Briefly, PLS calculates a covariance matrix between two (or more) sets of variables. [...] (pp. 40-42)

REVIEWER 1, COMMENT 3: I also wonder why “global signal variability” rather than “connectivity variability” or something similar is used here. The measure is not exactly based on the BOLD signal itself but rather on the correlations between two time series, i.e., connectivity.

*RESPONSE TO R1, COMMENT 3: We thank the reviewer for raising this point. While we agree that “connectivity variability” could have been a viable term for our covSTATIS-derived measures, it would have added to the confusion in terminology already present in the field of brain dynamics. By using the term “global signal variability” instead, we were able to unify two lines of work thus far treated independently in the literature: BOLD signal variability and dynamic functional connectivity. Building on the lay definition of the term “variability” (i.e., any measure of change), we decided to create a contrast in the spatial nature of our measures of interest by referring to BOLD signal variability as a *local* dynamic property and to dynamic functional connectivity as a *global* one, in line with previous work (Krohn et al., 2023 Science Advances). In doing so, we aimed at bringing more clarity to the field of BOLD dynamics, and at pushing researchers to jointly analyse *local* and *global* aspects of BOLD signal dynamics. We are now working on a separate manuscript that specifically aims to disambiguate the many possible ways of quantifying dynamical properties in BOLD time series. We expand on this in our response to the reviewer's comment 7.*

In this revision, we clarified our use of the word “global signal variability” and report changes below.

Revised results

Quantification of local and global BOLD signal variability

We first sought to identify robust regional metrics of local and global BOLD signal variability. By local and global BOLD signal variability, we refer to changes in the dynamic properties of single timeseries (local) and of pairs of timeseries (global). The former case is what is traditionally referred to as “BOLD signal variability” in the literature, and the latter as “dynamic functional connectivity”. In using the general term “variability” for both measures, we aimed at unifying two lines of work thus far treated independently yet shown to be inherently intertwined^{18,44}. (p. 5)

REVIEWER 1, COMMENT 4: Discussion: “This statistical explanation elucidates why, in this study, lower-order regions exhibited lower reliability of variability measures, than higher-order areas across fMRI samples.” It is quite confusing to come to this conclusion as all previous text on statistical

explanation is related to local and global BOLD signal variability (e.g., which one corresponds closer to the fMRI data etc). No lower-order and higher-order brain regions were mentioned before this. Please clarify and discuss.

RESPONSE TO R1, COMMENT 4: We thank the reviewer for bringing to our attention this source of confusion. We have now clarified the use of these terms in the results and discussion sections, specifically making a direct connection to the visionary work of Mesulam (Mesulam, 1998 Brain).

Revised Results

[...] For both local and global BOLD variability, reliability was lowest in sensori-motor (i.e., lower-order) areas and highest in heteromodal (i.e., higher-order), particularly default network, regions. (p. 9)

Revised Discussion

[...] Levels of statistical approximation are intuitively linked to levels of brain organization: lower levels of statistical approximation allow for more specific local neural representations **taking place in unimodal lower-order regions, while** higher levels of statistical approximation reflect lower-dimensional global neural representations **residing in heteromodal higher-order areas. Building on Mesulam’s visionary work¹¹⁴**, this statistical explanation elucidates why, in this study, lower-order regions exhibited lower reliability of variability measures, than higher-order areas across fMRI samples. (p. 25)

REVIEWER 1, COMMENT 5: What are the unique features of specific low-level and high-level brain regions in the context of local and global BOLD signal variability? The word “heterogenous” is used frequently but there is a lack of discussion on the patterns common or unique to local and global signal variability.

*RESPONSE TO R1, COMMENT 5: We thank the reviewer for raising this point. We purposefully chose not to discuss common/unique patterns of local and global BOLD signal variability, as setting up such contrast would require appropriate in-depth methodological testing that is outside of scope of the present work. Tracing common and unique local-global variability patterns would require extensive methodological analyses relating such patterns to multiple fMRI data types – to ensure reliability and generalisability, as our reliability analyses pointed out. It is for this reason that our Discussion section solely focuses on interpreting findings that are reliable and directly derived from our statistical approach (i.e., assessing local and global BOLD signal variability *independently*). We appreciate the reviewer’s interest, and we acknowledge the lack of clarity in our study’s goal. As such, we have now revised our Discussion section accordingly. Future work among the co-authors is seeking to determine such relations, and we hope that the present manuscript inspires such investigations as well.*

Revised Discussion

[...] Here, we showed how the statistical dependencies of local and global BOLD variability meaningfully interact with the statistical principles that govern the brain’s spatial organization: **cortical areas devoted to local processing presented variability patterns that were more dependent on fMRI data type for both local and global variability measures, than higher-order regions. These results call for more targeted empirical research on the complex interaction between local-global cortical processing, local-global BOLD signal variability and fMRI data type. While direct in-depth investigations on the commonalities and differences between low- and high-level cortical areas in the context of local and global BOLD variability are needed, here we provide a first general demonstration of the multiscale features of local and global BOLD signal variability, as independently assessed. Reliable local and global BOLD variability features are thus separately discussed here in turn.**

We found local and global BOLD variability to be spatially heterogeneous **multiscale** processes that closely map onto local and global information processes in the brain. [...] (pp. 25-26)

REVIEWER 1, COMMENT 6: It is nice to see a trend of how local and global BOLD variability relate to different gradients in young samples (Figure 3C). However, the level of evidence is a bit weak as some did not pass spin test in both young samples. Is there any reason behind this? For functional gradient, besides the principal unimodal-transmodal gradient, it might be informative to check the unimodal gradient (2nd one). Moreover, what is the pattern in the old sample? Do we see the same dissociation?

RESPONSE TO R1, COMMENT 6: Part 1 (sample differences): We agree with the reviewer that evidence on the multiscale mapping of local and global BOLD signal variability appears mixed, as spin test results are dependent on fMRI sample. We justify these mixed findings based on the coarse sample-dependent differences in the topography of local/global BOLD variability (highlighted in Figure 2 and Figure 3A-B). Throughout the paper, between-map correlations were run at the group level across brain regions: between-sample differences in correlation values are thus driven by differences in the respective maps' spatial distributions. Despite these inconsistencies, overall multiscale association patterns of local and global BOLD variability were largely maintained across samples, as shown by our Spearman's rho and dominance analysis results (Figure 3A bottom panel and Figure 3B). Together, while our work serves as a first-time comprehensive picture of BOLD variability's neurobiological features, we acknowledge that more research is needed to understand how such features relate to the brain's spatiotemporal architecture on an individual-by-individual basis. Resulting findings will be crucial to disentangle the sources of between-sample variance reported here. Following these considerations and comments, we have edited our manuscript accordingly and report changes below.

Part 1 (Sample Differences): Revised Discussion

[...] We found local and global BOLD variability to be spatially heterogeneous **multiscale** processes that closely map onto local and global information processes in the brain. **While some inconsistencies emerged across fMRI samples due to inter-sample differences in the topography of local and global BOLD signal variability, their multiscale properties were largely conserved across samples.** (p. 26)

[...] Despite the impact of fMRI acquisition sequences on local and global BOLD variability, the centrality of these measures in multiscale brain organization was reliable across fMRI datasets, further establishing their biological relevance. **More research is needed to understand how these multiscale features relate to the brain's spatiotemporal architecture on an individual-by-individual basis, given that our analyses were conducted at the group level and therefore disregarded intra-sample sources of variance.** (p. 28)

Part 2 (functional gradient 2): We thank the reviewer for suggesting considerations of more than the principal gradient. Following this suggestion, we probed the association between local and global BOLD signal variability with the second functional gradient. While the second gradient presented noisier, less reliable results in predicting local and global BOLD variability across samples, broad patterns of association were largely maintained across the two gradients. We provide these new results in-text and extensively report them in our Supplemental Materials. Edits are presented here below.

Part 2 (Functional Gradient 2): Revised Results

[...] As a final step, we repeated all analyses above including the second gradient of functional connectivity organization⁵⁴. Broad patterns of association were largely maintained across the two gradients, yet the principal gradient showed more reliable inter-sample results (**Figure S6**). (p. 18)

A BOLD variability association patterns integrating Margulies Gradient 2

B BOLD variability prediction patterns integrating Margulies Gradient 2

Local BOLD variability via rMSSD

Global BOLD variability via covSTATIS

Figure S6. Multiscale neurobiological correlates of local and global BOLD signal variability including the second functional gradient. | (A) Correlation matrices for each fMRI sample (upper triangle: Young Sample 1; lower triangle: Young Sample 2). Asterisks indicate correlations that survived significance testing (10,000 Hungarian spins of Schaefer's regional labels). We found positive associations between local BOLD signal variability and the second functional connectivity gradient ($r=0.54$, $p_{10k\ spin}=0.03$ Young Sample 1), and global BOLD signal variability and the second gradient ($r=0.70$, $p_{10k\ spin}<0.001$ Young Sample 1). As expected, participation coefficient scores of local and global BOLD variability including the second gradient largely overlapped with the values obtained by including only the first gradient (Young Sample 1: local = 0.62, global = 0.66; Young Sample 2: local = 0.49, global = 0.57). Additionally, in probing the inter-sample convergence of multiscale variability patterns, we found a significant inter-sample correlation only for local BOLD variability ($r=0.64$, $p=0.007$, 95% C.I. [0.22, 0.86]) and not for global BOLD variability ($r=0.49$, $p=0.05$, 95% C.I. [-0.001, 0.80]), differently from what we reported on the first gradient alone. | (B) Dominance analysis results per metric and fMRI sample including the second functional gradient.

Part 3 (Aging): We appreciate the reviewer's interest in understanding how multiscale patterns present themselves in older samples. We agree it is a valuable question for the field, especially given the number of investigations on age-related alterations in BOLD signal variability (e.g., Nomi et al., 2017; Boylan et al., 2020; Garrett et al., 2010; Garrett, Kovacevic, et al., 2013; Garrett et al., 2015, 2017, 2021; Goodman et al., 2024; Grady & Garrett, 2014; Millar, Ances, et al., 2020; Armbruster-Genç et al., 2016; Grady et al., 2023), and age-related differences in the gradient architecture of the brain (e.g., Bethlehem et al., 2022; Setton et al., 2023). However, a systematic and robust investigation on this topic would require individual-level data and analyses that, as mentioned in previous comments, are outside the scope of the present study.

REVIEWER 1, COMMENT 7: What is the additional value of the local and global BOLD signal variability proposed here on top of or in comparison with some other BOLD-based measures like static FC (the authors have found high correlation between static FC and global variability), ALFF, other DFC measures. It would be useful to demonstrate its behavioral relevance.

RESPONSE TO R1, COMMENT 7: We thank the reviewer for their enthusiasm and interest. These are fantastic questions that we (and others) are currently exploring in separate lines of work. Ultimately, integrating these projects in the current study would be outside of scope, as it would surpass the journal's word limit and additionally overload the reader, given how dense the paper is already. As such, we touch on these lines of work here below and give attention to this important issue in the discussion section.

First, we would like to address the issue of the metrics provided in the current manuscript. Overall, the value provided by our measures of local and global variability is their relatively straightforward implementation, their mathematical simplicity (i.e., rMSSD captures processes related to basic distributional properties of a timeseries), and user-independent nature. We expect our rMSSD findings to hold across other basic timeseries measures (i.e., standard deviation), and to relate to more complex measures given the slow, linear nature of fMRI (i.e., ALFF). We also expect our covSTATIS findings to be recapitulated by more standard sliding window approaches, given their relatively similar implementation.

Understanding inter-metric differences in BOLD variability is however a huge endeavour, as the field currently lacks a systematic framework that allows for inter-metric comparisons. To address this gap, in an ongoing project, we are putting forward a logically structured hierarchical grouping of BOLD variability measures. Such grouping organises all different metrics used in the literature in separate categories based on their mathematical implementation (distributional measures such as rMSSD used in this paper, linear – such as ALFF/fALFF – and non-linear measures, measures of stationarity and non-

stationarity). In doing so, our framework will help researchers better understand how existing methods fit in the broader literature, encouraging appropriate usage and interpretation of BOLD variability metrics (see **Draft Figure** below). In this context, inter-category comparisons, such as the one suggested by the reviewer, lose meaning if thought as comparisons between similar concepts: different categories instead capture different timeseries properties. This work was presented at OHBM 2025 in Brisbane (poster 1575, abstract available through Aperture Neuro here: <https://zenodo.org/records/15641972>).

As a technical side note to directly address the reviewer’s comment, (f)ALFF and variance (and therefore SD – the square root of variance – and rMSSD – $r > 0.9$ to SD on fMRI data) are highly similar in fMRI data. Mathematically speaking, ALFF measures are retrieved from the Autocorrelation Function (ACF). According to the Wiener-Khinchin theorem, the ACF contains all power spectral information of the data (stationary time series). As such, the area under the curve of the data’s power spectral density (PSD) equals to the data’s variance. If the data are filtered, as it is standard practice in fMRI, then the area under the PSD is mostly composed of low frequency fluctuations, precisely what ALFF is measuring.

To add to this theoretical work, we are also empirically testing for inter-metric differences in BOLD variability using synthetic timeseries. This work was presented at OHBM 2024. The abstract (#2415) is available on Aperture Neuro (Organization for Human Brain Mapping. Abstract Book 6: OHBM 2024 Annual Meeting. *Aperture Neuro*. 2024;4(Suppl 1)).

We would like to acknowledge the work of Wehrheim and colleagues on this topic as well (Wehrheim et al., 2024 Human Brain Mapping).

Draft Figure of the framework we are proposing in ongoing work, and we presented at OHBM 2025. This work is an in-prep manuscript, further highlighting the importance of the issue the reviewer raised.

Second, we would like to acknowledge the importance of the link between BOLD signal variability and behavior and appreciate the reviewer's interest in this relationship. There is a rich literature on the behavioral relevance of BOLD signal variability. In fact, the field of BOLD signal variability started from asking precisely these types of brain-behavior questions. Collectively, the field has demonstrated how BOLD variability provides unique behavioral information across a variety of trait (e.g., domains of cognition) and state (e.g., emotion) variables in healthy young adults, aging samples, and clinical populations. The first author of the present paper extensively reviews this literature in her PhD thesis, which has just recently been deposited and is accessible online (<https://www.proquest.com/openview/a8c1eeca7ba5c433c1caa187ff142c96/1?pqorigsite=gscholar&cbl=18750&diss=y>; Baracchini, 2024 McGill University). In the present submission, our goal was to provide a neurobiological framework to help contextualise, interpret, and guide existing and future behavioral research on the topic. We argue in favor of more investigations into BOLD variability's relationship with behavior.

Revised Discussion

[...] Finally, we acknowledge the wide array of brain signal variability metrics in the fMRI and MEG literatures^{18,103,145}, yet in the absence of a systematic framework for inter-metric comparisons, this study focused on two of the most popular signal variability metrics (i.e., rMSSD and dynamic functional connectivity). A logically structured grouping of variability measures is necessary to disambiguate the ontological confusion in the field and enable robust and interpretable inter-metric comparisons. Different variability metrics are derived from different mathematical formulations with inherently different assumptions: in other words, variability measures have different epistemologies and therefore they lead to distinct ontologies¹⁴⁶. In the presence of such framework, future research should test how the reported multiscale patterns of local and global brain signal variability may depend on the choice of variability metric.

Overall, the value provided by our measures of local and global variability is their relatively straightforward implementation, their mathematical simplicity (i.e., rMSSD captures processes related to basic distributional properties of a timeseries), and user-independent nature. We expect our rMSSD findings to hold across other basic timeseries measures (i.e., standard deviation), and to relate to more complex measures given the slow, linear nature of fMRI (i.e., (f)ALFF¹⁴⁷). We also expect our covSTATIS findings to be recapitulated by more standard sliding window approaches⁶⁴, given their relatively similar implementation. (p. 29)

Reviewer #2:

REVIEWER 2, SUMMARY: Key results: The manuscript examines the role of signal variability in brain function. Using several large fMRI datasets, the authors look at the association between variability and dynamic functional connectivity with microstructure, transcriptomics, neurotransmitter receptor and metabolic data, static connectivity, and empirical and simulated MEG data. They suggest that variability is structured and spatially heterogeneous and is important in multi-scale characterizations of brain function.

We thank the reviewer for their support for our study and their constructive feedback.

REVIEWER 2, COMMENT 1: Validity: I do not see any fatal flaws, but the reasoning behind methodological choices and variables included needs to be better described. These choices are especially important as they can significantly impact outcomes in variability work (the authors themselves suggest this). Specifically, in the literature, variability and dynamic functional connectivity (dFC) tend to be treated somewhat differently. I see no issues with choosing a dFC-type measure to index “global variability”, but the reasoning behind the choice needs to be framed in the context previous literature in the paper. In addition, local and global complexity has been discussed quite a bit with EEG/MEG (see for example <https://pubmed.ncbi.nlm.nih.gov/21525281/>; <https://pubmed.ncbi.nlm.nih.gov/23395850/>). I am not sure why these papers/ideas are not discussed as they are relevant, and why, in the context of MEG, they don't appear to be considered in analytic choices. Is this because the focus is more on variability than complexity in signal?

RESPONSE TO R2, COMMENT 1: We thank the reviewer for raising this important point and for bringing to our attention key references in this space. In this revision, we have better contextualised our choices of local and global BOLD signal variability metrics within the broader literature. We report changes below.

We additionally address the issue of complexity in this revision: in line with our response to reviewer 1's comments (see comment #7), we have expanded on the distinction between variability and other measures of signal dynamics, such as signal complexity. As correctly pointed out by the reviewer here, our research question focused on the biological nature of BOLD signal variability, and not of BOLD signal complexity. Measures of signal variability and signal complexity capture different aspects of a timeseries – whether it is on fMRI or MEG data, as these measures are derived from distinct mathematical formulations with inherently different assumptions (e.g., linearity vs non-linearity). The field currently lacks a systematic framework to allow for comparisons between these measures: in its absence, there remains significant ontological confusion around this topic. As such, we purposefully decided to solely focus on measures of signal variability (for both fMRI and MEG). However, in an ongoing project, we are putting forward a logically structured hierarchical grouping of time-series measures. Such grouping organizes all different metrics used in the BOLD signal variability literature in separate categories based on their mathematical implementation (distributional measures such as rMSSD used in this paper, linear – such as ALFF/fALFF – and non-linear measures, such as complexity measures). In doing so, our framework will help researchers better understand how existing methods fit in the broader literature, encouraging appropriate usage and interpretation of BOLD variability metrics. Inter-category comparisons, such as the one suggested by the reviewer, lose meaning if thought as comparisons between similar concepts: different categories instead capture different timeseries properties. This work was presented at OHBM 2025 in Brisbane (poster 1575, abstract available through Aperture Neuro here: <https://zenodo.org/records/15641972>). As such, we touch on these lines of work here below and give attention to this important issue in the discussion section.

Revised Introduction

[...] Second, human brain function is organized along hierarchical modules, ranging from local functional units to global functional networks^{35,36}, with heterogeneous topographies. Such organization enables information to be both segregated (local) and integrated (global) across units, subserving different neural and cognitive functions – as observed in various measures such as mean functional activation, functional connectivity, computational and BOLD signal complexity studies³⁷⁻⁴². Along these lines, BOLD signal variability must therefore also be understood within a local-global framework^{43,44} that considers both the methodology and the analysis level of interest. While previous fMRI reports have investigated the local-global nature of BOLD signal variability, they have generally focused on *static* global measures of brain organisation (i.e., the correlation between local mean BOLD signals over time, *functional connectivity*⁴⁵) and more local characterization of variability on a single scale (e.g., voxel or region). The temporal variability of individual regions (local BOLD signal variability) must thus be evaluated alongside the *dynamic* coordination of functional neural units (dynamic functional connectivity, here global BOLD signal variability). If biologically relevant, local and global BOLD signal variability should present spatially heterogeneous topographies that recapitulate known neurobiological processes that unfold across spatial scales. (p. 4)

Revised results

Quantification of local and global BOLD signal variability

We first sought to identify robust regional metrics of local and global BOLD signal variability. By local and global BOLD signal variability, we refer to changes in the dynamic properties of single timeseries (local) and of pairs of timeseries (global). The former case is what is traditionally referred to as “BOLD signal variability” in the literature, and the latter as “dynamic functional connectivity”. In using the general term “variability” for both measures, we aimed at unifying two lines of work thus far treated independently yet shown to be inherently intertwined^{18,44}. (p. 5)

Revised Discussion

[...] Finally, we acknowledge the wide array of brain signal variability metrics in the fMRI and MEG literatures^{18,103,145}, yet in the absence of a systematic framework for inter-metric comparisons, this study focused on two of the most popular signal variability metrics (i.e., rMSSD and dynamic functional connectivity). A logically structured grouping of variability measures is necessary to disambiguate the ontological confusion in the field and enable robust and interpretable inter-metric comparisons. Different variability metrics are derived from different mathematical formulations with inherently different assumptions: in other words, variability measures have different epistemologies and therefore they lead to distinct ontologies¹⁴⁶. In the presence of such framework, future research should test how the reported multiscale patterns of local and global brain signal variability may depend on the choice of variability metric.

Overall, the value provided by our measures of local and global variability is their relatively straightforward implementation, their mathematical simplicity (i.e., rMSSD captures processes related to basic distributional properties of a timeseries), and user-independent nature. We expect our rMSSD findings to hold across other basic timeseries measures (i.e., standard deviation), and to relate to more complex measures given the slow, linear nature of fMRI (i.e., (f)ALFF¹⁴⁷). We also expect our

covSTATIS findings to be recapitulated by more standard sliding window approaches⁶⁴, given their relatively similar implementation. (p. 29)

REVIEWER 2, COMMENT 2: Originality and significance: Several papers have examined the importance/unique contribution of variability measures in understanding brain function (seminal papers

include <https://www.pnas.org/doi/full/10.1073/pnas.0901831106>; <https://www.jneurosci.org/content/30/14/4914>; <https://www.ncbi.nlm.nih.gov/pmc/articles/PMC3922711/>). The value added in this manuscript is the impressive amount of data examined, the gamut of micro and macroscale variables that brain signal variability was linked to, and the examination of local and global variability in BOLD fMRI. To better highlight significance, the authors should focus on these unique contributions rather than the idea that variability is a useful metric. They should better motivate their choice of macro and microscale variables, their analysis methods (some which do not follow directly from previous literature), and include greater discussion of the implications of their specific findings (i.e., why and how is brain signal variability an important measure for brain characterisation? How can it fit with / add to other more commonly used measures? How should knowing that variability has “spatially heterogeneous topography, encapsulate micro-, meso- and macroscale neurobiological phenomena” influence the use of this measure in cognitive network neuroscience?).

RESPONSE TO R2, COMMENT 2: We thank the reviewer for their valuable feedback and detailed suggestions. While we agree with the reviewer that other important studies have examined the functional relevance of brain signal variability, its characterisation in fMRI remains limited. Existing fMRI reports have shown relationships between local BOLD variability and local BOLD activity (Garrett et al., 2011), a region’s local variability and its whole brain functional connectivity (Garrett et al., 2018), local variability and non-grey matter tissue (Millar, Petersen, et al., 2020), and local variability and system-level dopaminergic and GABAergic availability (Garrett et al., 2015; Lalwani et al., 2021). Most of these studies however are carried out in aging populations, underscoring the need for broader examinations in younger samples. Furthermore, while the majority of BOLD signal variability studies focus on variability’s behavioural, cognitive, and clinical correlates, these associations do not directly establish its role in brain organisation.

Following this reviewer's suggestion, in this revision and provided below, we revised the manuscript to acknowledge the key issues raised regarding our study design, and we have included greater discussion on the significance and implications of our findings. We acknowledge that this issue was raised by both reviewers, and believe the current revision appropriately contextualizes the current findings.

Revised Introduction

[...] The human brain is a complex system. From the processing of incoming information to the generation of motor outputs, variability is present at all levels of the central nervous system^{6,7}. Brain activity varies across multiple timescales, from milliseconds to years, and these temporal changes can be observed across multiple spatial scales, from regions to networks^{8,9}. Despite its pervasiveness, there continues to be resistance to exploring variability in human cognitive neuroscience. Human behavior has been shown to be stochastic^{7,10,11} and computational models have operationalized the brain as a complex dynamical system¹²⁻¹⁶, yet corresponding empirical neuroimaging research still lags behind. (p. 3)

[...] Thus far, BOLD signal variability has been primarily studied in relation to behavior, cognition, development, and clinical status¹⁸⁻²⁴. Demonstrating such associations does not, however, directly establish BOLD signal variability’s biological role in whole-brain organization. As such, it remains

unclear whether and to what extent BOLD signal variability investigations are capturing meaningful neural signatures or system noise. To this end, a few recent studies have begun to characterize BOLD variability's neurobiological underpinnings, yet these investigations are typically restricted to comparisons across age and focus on single, mostly macroscale, neurobiological features²⁵⁻²⁹. A broader and more integrated assessment of BOLD signal variability in younger samples is therefore needed, one that examines its statistical, topographical and neuronal properties and systematically relates BOLD signal variability to a wider array of neurobiological features. Evidence in favour of taking such an integrative multi-scale approach to BOLD signal variability comes from multiple lines of empirical and computational work collectively demonstrating the role of cellular, molecular, genetic, and metabolic factors in shaping local and global hemodynamic signals³⁰⁻³⁴. (pp. 3-4)

[...] To understand the spatial organization and biological properties of local and global BOLD variability, inspired by recent multiscale fMRI investigations^{32,46-49}, we examined their topography within each fMRI dataset and interrogated associations with open-source data, including *ex_vivo* histology⁵⁰ and *in_vivo* microstructure⁵¹, transcriptomics⁵²⁻⁵³, PET-derived neurotransmitter receptor and metabolic information⁴⁷, and fMRI static connectivity data^{54,55}. (pp. 4-5)

Revised Discussion

Response to [Why and how is brain signal variability an important measure for brain characterisation?]

[...] Brain signal variability has been increasingly recognized as a fundamental feature of neural functioning. At the cellular and molecular level, studies have shown how neural variability serves as preparatory activity for future stimulus processing. It allows for the recapitulation of previous sensory experience⁹⁷⁻⁹⁹, predicts ongoing behavioral and cognitive states¹⁰⁰⁻¹⁰⁴, facilitates synaptic formation^{100,105-107} and drives signal propagation¹⁰⁸. At the macroscale level in fMRI, accumulating evidence has highlighted BOLD signal variability as a key feature of optimal behavioral functioning and a marker of neurocognitive aging and clinical status^{29,21,25-27,29,92-95,109-113}. Our work builds on these multiple lines of evidence and further expands the relevance of brain signal variability from a multiscale neurobiological standpoint. (pp. 24-25)

Response to [How can it fit with / add to other more commonly used measures?]

[...] Overall, the value provided by our measures of local and global variability is their relatively straightforward implementation, their mathematical simplicity (i.e., rMSSD captures processes related to basic distributional properties of a timeseries), and user-independent nature. We expect our rMSSD findings to hold across other basic timeseries measures (i.e., standard deviation), and to relate to more complex measures given the slow, linear nature of fMRI (i.e., (f)ALFF¹⁴⁷). We also expect our covSTATIS findings to be recapitulated by more standard sliding window approaches⁶⁴, given their relatively similar implementation. (p. 29)

Response to [How should knowing that variability has “spatially heterogeneous topography, encapsulate micro-, meso- and macroscale neurobiological phenomena” influence the use of this measure in cognitive network neuroscience?]

[...] By establishing BOLD signal variability as a spatially heterogeneous, multifactorial, multimodal property of brain organization integral to healthy brain function, this study underscores the value of systems-level approaches to brain function¹⁴⁸. Our work provides fertile soil for new theoretical and methodological work in cognitive network neuroscience, by opening the door to neurobiologically

grounded hypotheses on the role of local and global BOLD signal variability in cognition and behavior. Moreover, our findings offer empirically grounded targets for generative models of brain signal dynamics in health and disease. (p. 30)

REVIEWER 2, COMMENT 3: Data & methodology: The data and methodology are sufficiently described in text (with references to more detailed previously published descriptions) to enable reproducing the results. As a minor point, added details regarding Group Prior Individual Parcellation (GPIP), specifically how it may influence results over more commonly used parcellation methods would be useful.

RESPONSE TO R2, COMMENT 3: We thank the reviewer for their positive feedback on our data and methodology. We appreciate their suggestion to provide more detail on the GPIP approach, and we have now clarified this in the revised manuscript. Edits are provided here below.

Briefly, the goal of our work was to understand how neural signal dynamics contribute to the rich spatial architecture of the human brain. As such, we wanted to ensure high topographical precision of fMRI BOLD. Individualised parcellation approaches, including GPIP, precisely enable this. While standard group-level parcellation methods are more commonly used in fMRI research, they often yield coarse and imprecise estimations of individual-level regional boundaries and therefore of the spatial unfolding of fMRI BOLD. Compared to standard group-level parcellation methods, individualised parcellation approaches offer substantially improved sensitivity to individual differences in brain activity and behaviour, by adjusting each individual's regional boundaries to their own functional connectivity organisation (Chong et al., 2017 NeuroImage; Kong et al., 2018 Cerebral Cortex; Levi et al., 2023 Network Neuroscience; Sassenberg et al., 2023 NeuroImage). To note, we selected GPIP as our parcellation method of choice also for consistency with previous work from our group using the same data (Setton et al., 2023 Cerebral Cortex).

Revised Methods

fMRI Young Sample 1

[...] Functional images were parcellated using the Group Prior Individual Parcellation (GPIP)^{69,149}, a participant-specific parcellation approach initialized on the Schaefer 200-17 network solution⁵⁶. We selected GPIP as our parcellation method of choice to ensure high topographical precision of fMRI BOLD, and for consistency with previous work from our group using the same data⁶⁹. Unlike standard parcellations, GPIP accounts for within-subject variation in parcel boundaries, offering substantially improved sensitivity to individual differences in brain activity and behaviour¹⁴⁹⁻¹⁵². Specifically, GPIP consists of an iterative Bayesian process: it starts from the user-provided reference group atlas (Schaefer 200-17 in our case) and iteratively refines each individual's parcel boundaries relative to their resting-state connectivity data¹⁴⁹. It then computes concentration matrices (inverse covariance/partial correlation) for the whole sample via a group sparsity constraint. To ensure stability and precision, the algorithm iterates between the former and latter step for a total of 20 iterations or until no more than one vertex is changing per parcel¹⁴⁹. For more details about GPIP, the reader is referred to our previous publication where we used GPIP on this sample⁶⁹. (pp. 31-32)

REVIEWER 2, COMMENT 4: Appropriate use of statistics and treatment of uncertainties: The statistical tests appear to be performed appropriately.

RESPONSE TO R2, COMMENT 4: We thank the reviewer for their positive evaluation of our statistical approach.

REVIEWER 2, COMMENT 5: Conclusions: *Data interpretation and related conclusions are valid.*

RESPONSE TO R2, COMMENT 5: We appreciate the reviewer's positive feedback on our conclusions.

REVIEWER 2, COMMENT 6: Suggested improvements: *No additional data or experiments are necessary.*

RESPONSE TO R2, COMMENT 6: We thank the reviewer for their positive assessment of our study.

REVIEWER 2, COMMENT 7: References: *Some of the references mentioned above need to be added. In addition:*

o The following sentence is missing references "The human brain is a complex stochastic system, hence integrating functional variability of brain signals is essential for the modeling of human brain function."

Deco et al's (2009) paper from above speaks to this and could be added.

o The authors suggest that "computational approaches to brain dynamics may thus benefit from introducing spatial heterogeneity in local measures of variability, to more precisely approximate empirical observations." Has this not been done? And if not, this is really interesting and would benefit from a little more discussion.

o The finding that global variability increases with age could be better contextualised within the literature, as I believe this is the area in which fMRI signal variability analyses are most commonly performed. There variability papers already cited in the manuscript show consensus with overall increases in variability with age, but an acknowledgement that there are some papers that found general decreases or a mix of increases and decreases (e.g., <https://pubmed.ncbi.nlm.nih.gov/32915200/>; <https://pubmed.ncbi.nlm.nih.gov/35901557/>) could be included.

RESPONSE TO R2, COMMENT 7: We thank the reviewer for their suggestions and comments.

[Point 1]: We agree with the reviewer and have now incorporated the suggested reference in our revised manuscript.

Revised Discussion

[...] The human brain is a complex stochastic system¹⁶, hence integrating functional variability of brain signals is essential for the modeling of human brain function. (p. 24)

[Point 2]: We thank the reviewer for their insightful comment. We agree that our sentence on computational approaches to local dynamics would benefit from a bit more discussion.

Computational modelers typically favor model parsimony, that is model simplicity and tractability, over high biological precision. For this reason, early population models assumed and enforced spatial homogeneity in local dynamics (Wilson and Cowan, 1972 *Biophysical Journal*; Jansen and Rit, 1995 *Biological Cybernetics*; Robinson et al., 1997 *Physical Review E*; Jirsa and Haken, 1997 *Physica D: Nonlinear Phenomena*). Only in recent years, modelers have begun integrating spatial heterogeneity – from receptor density, gene expression, myelination, cortico-thalamic functional gradient maps – in their models, to more accurately capture key features of brain dynamics observed empirically, including

covariance, dynamic functional connectivity, regional times-scales, and stimulus sensitivities (Deco et al., 2013 Journal of Neuroscience; Breakspear, 2017 Nature Neuroscience; Wang et al., 2020 Nature Review Neuroscience; Herzog et al., 2024 Network Neuroscience; Froudust-Walsh et al., 2021 Neuron; Klatzmann et al., 2025 Cell Reports; Müller et al., 2023 Cell Reports; Müller et al., 2020 Nature Communications). It is in this context that we argue that our results will help shape existing and future neurobiologically-informed computational models, by further constraining them with a new spatially heterogeneous axis of brain dynamics.

In response to the reviewer's suggestion, we have now expanded our discussion to clarify our point and contextualise our argument in relation to past and current modeling approaches.

Revised Discussion

[...] Our study adds to this body of evidence by revealing that local BOLD variability is a structured, spatially heterogeneous process. Computational approaches to brain dynamics may thus benefit from introducing spatial heterogeneity in local measures of variability, to more precisely approximate empirical observations. In doing so, empirical fMRI data will not only serve as a model validation tool but also as a model optimization tool. While early population models assumed and enforced spatial homogeneity in local dynamics¹²⁰⁻¹²³, recent years have progressively seen the integration of spatial heterogeneity – from receptor density, gene expression, myelination, cortico-thalamic functional gradient maps – in neural mass models, to more accurately capture key features of brain dynamics observed empirically, including covariance, dynamic functional connectivity, regional times-scales, and stimulus sensitivities^{14,15,34,124-128}. It is in this context that we argue that our results will help shape existing and future neurobiologically-informed computational models, by further constraining them with a new spatially heterogeneous axis of brain dynamics. (p. 26)

[Point 3]: We thank the reviewer for their suggestion. In our manuscript, we found age-related decreases (and not increases) in covSTATIS-derived global signal variability, in line with most BOLD signal variability studies in the literature. The reviewer is correct in saying that BOLD variability investigations are commonly carried out in aging populations, as aging researchers were among the first to investigate BOLD signal variability as a potential meaningful construct in fMRI (e.g., Randy McIntosh, Cheryl Grady and Douglas Garrett). In this revision, in line with the reviewer's suggestion, we have briefly contextualized our aging-related findings. As these results primarily served as a validation for the covSTATIS method, their contextualization remains concise in our revised discussion.

Revised Discussion

[...] Local and global BOLD signal variability are not merely sources of biologically irrelevant noise. They are reduced with age in line with most prior investigations^{19-21,25-27,29,92-94}, though some studies have reported opposite effects^{95,96}. We have shown how measures of BOLD signal variability have a spatially heterogeneous topography, they encapsulate micro-, meso- and macro-scale neurobiological phenomena, and are related to underlying electrophysiological neuronal activity. (p. 24)

REVIEWER 2, COMMENT 8: Clarity and context: The abstract is clear and accessible but the introduction and discussion are overly concise, especially for an experimental design as complex as the one used in this paper.

RESPONSE TO R2, COMMENT 8: We thank the reviewer for their feedback. In responding to all of their comments and to the comments of reviewer 1, we have expanded the introduction and discussion

sections of our paper to better reflect the complexity of our experimental design and to provide additional context for our results. We aimed to strike a balance between general audience accessibility and scientific fidelity. We appreciate both reviewers for their insight and comments.

REVIEWER COMMENTS

Reviewer #1 (Remarks to the Author):

I appreciate the hard work from the authors. The revision has addressed some concerns, but there are still three main concerns left:

REVIEWER 1, COMMENT 1: What are the unique features of specific low-level and high-level brain regions in the context of local and global BOLD signal variability? The authors did not perform any additional analyses to compare them quantitatively. I think this is important even though they are derived independently. The newly added discussion is also non-specific and not satisfying. E.g. “cortical areas devoted to local processing presented variability patterns that were more dependent on fMRI data type for both local and global variability measures, than higher-order regions.” What are the cortical areas devoted to local processing? How is it different from higher-order regions? Given the rich literature on functional gradients and organization involving sensorimotor and association regions, more comparison and interpretation for local and global variability is needed. To me, this response contradicts the authors’ claim in their earlier response that these two processes are linked, and in this study, they purposely named them differently from the literature to study both (single time-series variability and paired time series (connectivity) variability). Without fully addressing this point, providing additional insights for the field is hard.

We thank the reviewer for their constructive feedback. We agree that quantitatively comparing low-level and high-level brain regions in the context of local-global signal variability metrics is an important direction for future work. Yet, the present study did not seek to directly address this question. We apologize if our aims were not clear in our earlier response and have revised our manuscript to clarify our goals accordingly. See below.

Importantly, performing additional quantitative analyses to contrast local-global variability would significantly blow out an already extensive and comprehensive assessment of variability and fall outside the scope of this paper. We are actively pursuing this investigation in a separate project.

Lastly, we respectfully disagree with the reviewer that failure to quantitatively address their request would undermine the impact of our work. As outlined in our manuscript, the field of BOLD signal variability remains significantly detached from its neurobiological and statistical underpinnings. Our work directly addresses this gap and provides a conceptual foundation for future studies to build upon.

Revised Results (note that the abstract, revised introduction and revised discussion already state that local and global variability are treated independently):

[...] We first sought to identify robust regional metrics of local and global BOLD signal variability. By local and global BOLD signal variability, we refer to changes in the dynamic properties of single timeseries (local) and of pairs of timeseries (global). The former case is what is traditionally referred to as “BOLD signal variability” in the literature, and the latter as “dynamic functional connectivity”. In using the general term “variability” for both measures, we aimed at **conceptually juxtaposing two lines of research that have traditionally been treated independently but recently shown to be inherently intertwined^{18,44}. In doing so, our work establishes the foundation for future studies to quantitatively compare them. (p. 5)**

REVIEWER 1, COMMENT 2: Patterns in the sample with older adults. This question has not been addressed. I do not think it is unreasonable to repeat the same analyses in old samples.

We agree with the reviewer that investigations into aging and lifespan development are of particular interest. Yet, they are out of scope of the present manuscript, which focuses primarily on younger adults and already entails a significant number of analyses. A proper treatment of aging would require an extensive set of additional analyses, as aging affects both neural signal properties and the underlying neurobiology. In sum, repeating our analyses in an aging cohort would result in a separate line of inquiry rather than a direct extension of this study and ensuing findings would warrant substantial interpretation and discussion that extend well beyond the word limit of this manuscript.

REVIEWER 1, COMMENT 3: Regarding my last comment on the relationship and comparisons with other measures, I am aligned with Reviewer 2 R2 that it is important to demonstrate why this metric is better than others and their implications in understanding our brain development, aging and behavior associations.

We thank the reviewer for rehearsing this point. Reviewer 2 was satisfied with how we addressed this specific comment. As discussed in our previous response and in the manuscript, different measures of signal variability and complexity capture conceptually different timeseries properties. We recognize the importance of benchmarking these approaches, yet, doing so would significantly shift the focus of the current work and compromise its clarity and coherence. As noted previously, we are actively investigating this in a separate, dedicated project.

Reviewer #2 (Remarks to the Author):

The authors have addressed all my concerns.

We thank the reviewer for their constructive comments.